# Tropospheric mixing and parametrization of unresolved convective updrafts as implemented into the Chemical Lagrangian Model of the Stratosphere (CLaMS v2.0)

Paul Konopka, Mengchu Tao, Felix Ploeger, Mohamadou Diallo, and Martin Riese

Forschungszentrum Jülich, (IEK-7), Germany

*Correspondence to:* Paul Konopka (p.konopka@fz-juelich.de)

**Abstract.** Inaccurate representation of mixing in chemistry transport models, mainly suffering from an excessive numerical diffusion, strongly influences the quantitative estimates of the stratosphere-troposphere exchange (STE). The Lagrangian view of transport offers an alternative to exploit the numerical diffusion for parametrization of the physical mixing. Here, we follow this concept and discuss how to extend the representation of tropospheric transport in the Chemical Lagrangian Model of the
Stratosphere (CLaMS).

Although the current transport scheme in CLaMS (v1.0) shows good ability of representing transport of tracers in the stably stratified stratosphere (Pommrich et al. (2014) and the references therein), there are deficiencies in representation of the effects of convective uplift and mixing due to weak vertical stability in the troposphere. We show how the CLaMS transport scheme was modified by including additional tropospheric mixing and vertical transport due to unresolved convective updrafts by
parametrizing these processes in terms of the dry and moist Brunt-Vaisala frequency, respectively. The regions with enhanced convective updrafts in the novel CLaMS simulation covering the 2005-08 period coincide with regions of enhanced convection as diagnosed from the satellite observations of the Outgoing Longwave Radiation (OLR) .

We analyze how well this approach improves the CLaMS representation of $CO_2$ in the upper troposphere and lower stratosphere, in particular the propagation of the $CO_2$ seasonal cycle from the Planetary Boundary Layer (PBL) into the lower
stratosphere. The $CO_2$ values in the PBL are specified by the CarbonTracker data set (version CT2013B) and the Comprehensive Observation Network for TRace gases by AIrLiner (CONTRAIL) observations are used to validate the model. The proposed extension of tropospheric transport increases the influence of the PBL in the middle and upper troposphere and at the same time impacts the STE. The effect on mean age away from the troposphere in the deep stratosphere is weak.

## 1   Introduction

Modeling of transport from a Lagrangian perspective has gained increasing popularity in the last few decades not only within the atmospheric community. The chance to avoid, or at least to minimize, the numerical diffusion ever present in Eulerian numerical schemes is the strongest motivation for the Lagrangian formulation of transport. Despite the obvious advantage of the Lagrangian view separating mixing from the advective part of transport, only very few Lagrangian chemical transport

models (CTMs) with explicit mixing exist so far (e.g. Collins et al., 1997; Fairlie et al., 1999; Reithmeier and Sausen, 2002; McKenna et al., 2002b; Konopka et al., 2007; Wohltmann and Rex, 2009; Pugh et al., 2012).

In this paper, the 3D version of the Chemical Lagrangian Model of the Stratosphere (CLaMS) will be used (McKenna et al., 2002b, a; Konopka et al., 2004). The novel approach of CLaMS is its parametrization of atmospheric mixing, especially in the stratosphere where vertical mixing is extremely weak due to a strong vertical stability. Whereas the common approach is to
minimize the numerical diffusion ever present in the modeling of transport, CLaMS is a first attempt to apply this "undesirable disturbing effect" to parametrize the "true" physical mixing. This idea is realized by using some scaling properties of numerical diffusion, which are the same as of the atmospheric diffusivity (Konopka et al., 2007, 2012; Pommrich et al., 2014) and by applying numerical regridding only to the strongly deformed parts of the Lagrangian grid where physical mixing is expected anyway (McKenna et al., 2002b; Konopka et al., 2004). This novel parametrization of mixing was included into the well-known
pure Lagrangian, i.e., trajectory-based representation of transport (e.g. Stohl et al., 2005; Legras et al., 2005; Bowman et al., 2007; Wernli and Davies, 1997).

However, in the current version of CLaMS v1.0 (Pommrich et al., 2014) only deformations within quasi-isentropic layers driven by horizontal strain and vertical shear rates are taken into account. This is certainly a good approximation in a stably stratified stratosphere suppressing vertical mixing but not in the troposphere where vertical mixing is expected. To extend the
CLaMS idea of "sufficiently strong", almost isentropic deformations triggering mixing to the troposphere we use the concept of atmospheric stability. The flow is called unstable if a small perturbation at initial time will exponentially grow during the course of the evolution of the flow.

One of the widely used parameters quantifying instabilities is the gradient Richardson number, $Ri$ describing the onset of instabilities driven by wind shear and/or buoyancy (e.g. Turner, 1973; Stull, 1988). $Ri$ is defined as $Ri = N^2/[(du/dz)^2 +$
$(dv/dz)^2]$ with $du/dz$, $dv/dz$ denoting the vertical shear of the horizontal wind components. $N$ is the (dry) Brunt-Vaisala frequency quantifying buoyancy-driven turbulence in terms of the potential temperature lapse rate, i.e. $N^2 = (g/\theta)d\theta/dz$ ($\theta$ - potential temperature, $g$ - gravity of Earth and $z$ geometric altitude). The flow becomes dynamically unstable or even turbulent when $Ri < Ri_c$ ($Ri_c$ critical value of $Ri$). This occurs either when the wind shear is strong enough to outweigh any stabilizing buoyant forces (denominator in the definition of $Ri$ is large), or when the dry or, more general, moist environment is statically
unstable (numerator in the definition of $Ri$ is small or even negative because the lapse rate $d\theta/dz$ is small or even negative). The scale-dependent value of $Ri_c$, is about 0.25 although reported values have ranged from roughly 0.2 to 1.0 (Balsley et al., 2008).

In the stratosphere, where the flow is characterized by high static stability, only almost isentropic deformations driven by the horizontal strain and vertical shear are considered in the current version of CLaMS v1.0 (Pommrich et al. (2014) and the refer-
ences therein). These deformations measured in terms of the Lyapunov exponent $\lambda$ are used in CLaMS to parametrize mixing within layers, which are parallel to the isentropes about the level of 300 hPa. However, the effect of vertical instabilities being a dominant feature of tropospheric transport is not taken into account. To parametrize such (potentially) vertically unstable regions by using "sufficiently small" values of the dry or moist Brunt-Vaisala frequency $N$ or $N_m$ is our main heuristic idea,

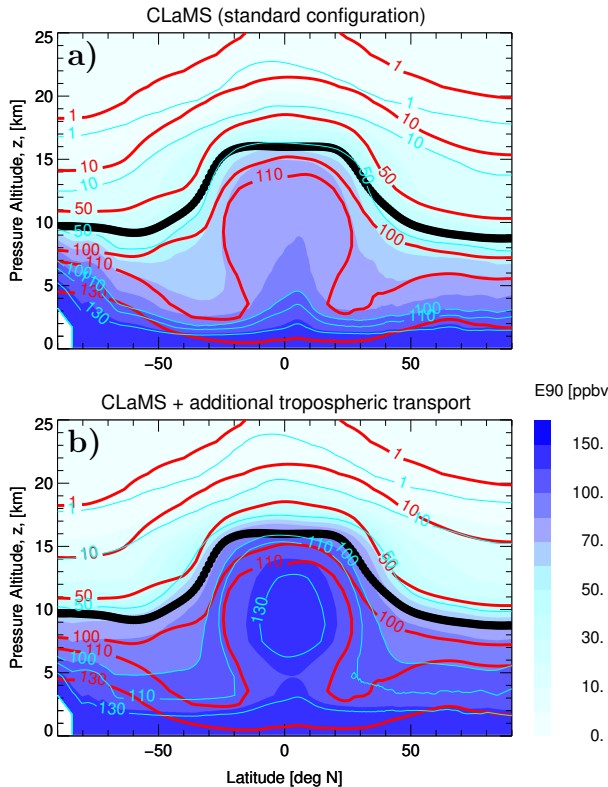

**Figure 1.** The different shades of blue quantify the annual mean of the e90 tracer mixing ratios (calculated for 2007) as a function of latitude and log-pressure altitude and are derived from the current version of CLaMS v1.0 described in Pommrich et al. (2014) (a) and from the here discussed version with extended tropospheric transport (b). For comparison, the WACCM 1955-2099 climatology of e90 is also shown (red isolines) as described in Abalos et al. (2017). The mean tropopause is depicted by the black line.

how to extend the CLaMS transport scheme from the deformation-driven to the deformation- *and* vertical instability-driven scheme.

Generally, CLaMS tropospheric tracers like CO or $CH_4$ show reasonable mixing ratios in the Upper Troposphere and Lower Stratosphere (UTLS) only if their values in the orography-following lowest model boundary (thickness: 1.1-1.3 km) are artificially enhanced by a factor 1.5-2 (Pommrich et al., 2014). The consequence of this enhancement are too strong vertical gradients of these tracers, especially in the lower and middle troposphere if compared with observations (not shown). A possible reason might be that a significant part of the upward vertical transport within the troposphere is underestimated.

Figure 1 shows a comparison in terms of the e90 tracer between the current CLaMS version (top), the here discussed extension of tropospheric transport (bottom) with the Whole Atmospheric Community Climate Model (WACCM, red isolines) which is known for a good representation of tropospheric transport and chemistry (see e.g. Park et al. (2013) and the references

therein). Here, the CLaMS annual means (calculated for 2007) are compared with the WACCM 1955-2099 climatology derived from the REF-C2 reference simulations which have coupled atmosphere, ocean, ice, and land components (Abalos et al., 2017) (year-to-year variability in WACCM climatology is much smaller if compared with deviation of CLaMS from the WACCM climatology.

Similar like in WACCM, the artificial e90 tracer, with a constant e-folding lifetime of 90 days, is set to 150 ppb everywhere in the lowest layer of CLaMS. The e90 tracer is suitable to diagnose typical timescale of transport from the Planetary Boundary Layer (PBL) into the lower stratosphere (Prather et al., 2011; Abalos et al., 2017). Figure 1 shows that CLaMS in the current version significantly underestimates the upward transport if compared with the WACCM model and that this comparison improves if the new version of transport is included.

In this paper, we aim to parametrize the unresolved processes like convective updrafts and tropospheric mixing whose representation in global reanalysis data is uncertain (Russo et al., 2011). Although more complex convective schemes exist both in the Eulerian (e.g. Tiedtke (1989); Emanuel (1991)) as well as in the Lagrangian formulation (e.g. Collins et al. (2002); Erukhimova and Bowman (2006); Forster et al. (2007); Pugh et al. (2012); Ueyama et al. (2018); Brinkop and Jöckel (2018)), our approach mainly intends to cover the range of possible variability due to unresolved tropospheric transport. Guided by diabatic rather than kinematic thinking and by the wish to reduce the complexity of convection as much as possible, we aim here to create a technical framework which allows us in the future to estimate the impact of such uncertainties on the composition of the UTLS as well as on the Stratosphere-Troposphere-Exchange (STE). Even small changes in the concentration and distribution of radiatively active gases in the UTLS such as water vapor or ozone significantly impact radiative forcing on surface temperature (Riese et al. (2012), IPCC2014).

In the next section, we describe the properties and mean distributions of the dry or moist Brunt-Vaisala frequency $N$ or $N_m$ as derived from the meteorological data. $N$ and $N_m$ are used in this paper to parametrize the additional tropospheric transport. Section 3 explains technical details of this parametrization. Section 4 describes the CLaMS setup (v2.0), details of the performed model simulations as well as some results diagnosing which regions of the atmosphere are mainly affected by our extension of transport. To validate these new properties, we discuss the transport of $CO_2$ from the PBL into the lower stratosphere and compare the respective distributions with the airborne observations and the CarbonTracker model simulations. Finally, we discuss our results in section 5.

## 2   Vertically unstable troposphere versus stably stratified stratosphere

Static stability can be quantified in terms of the (dry) Brunt-Vaisala frequency (BVF) via $N^2 = (g/\theta)(d\theta/dz)$. $N$ describes the frequency at which an air parcel oscillates when displaced vertically in a statically stable environment, i.e. within a region with a positive lapse rate $d\theta/dz > 0$ (for some details see appendix A). Because the well-mixed troposphere is characterized by low values of $N$ and the stably stratified stratosphere by high values of $N$, it is expected that this difference also manifests in the corresponding vertical diffusivities (large and small for the troposphere and the stratosphere, respectively).

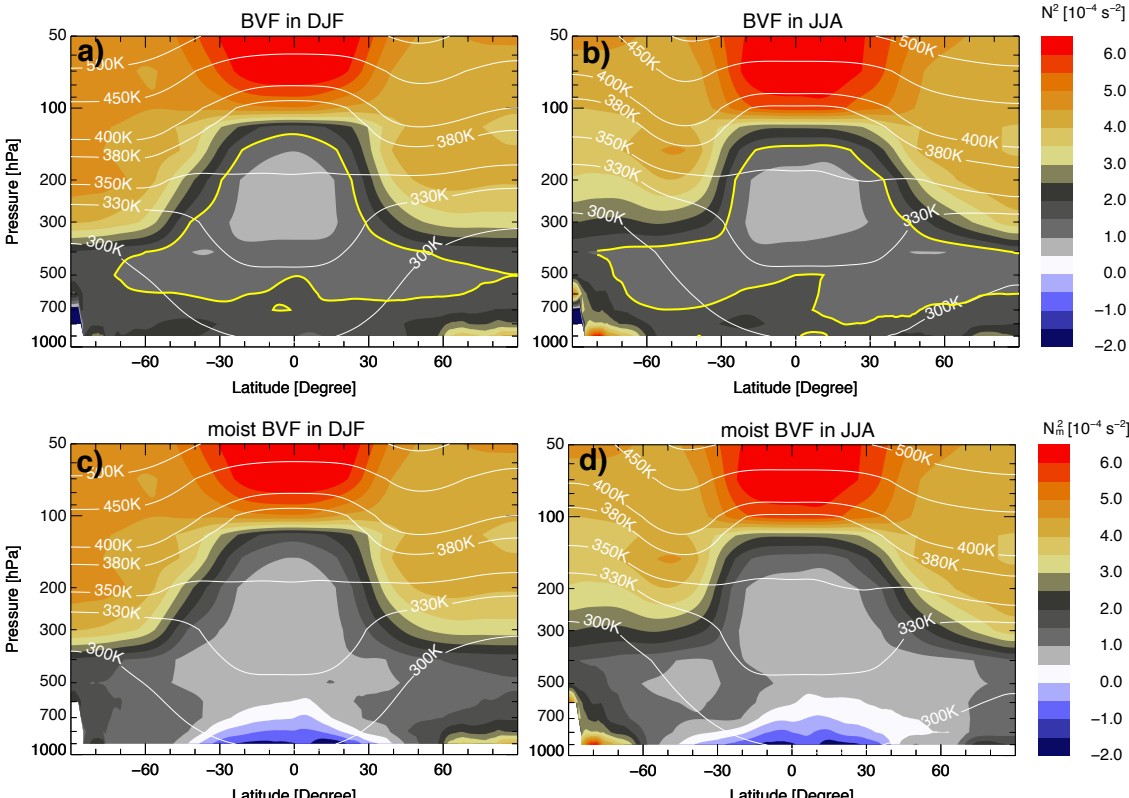

**Figure 2.** The zonal mean distribution of dry, $N^2$ (top) and moist, $N_m^2$ (bottom) Brunt-Vaisala frequency (BVF) in the troposphere and the lower stratosphere for DJF (a, c) and JJA (b, d) as calculated from 2005 ERA-Interim data. The thick yellow contours in the top row highlight those part of the atmosphere where the tropospheric mixing is included into CLaMS. The bluish colored regions in the bottom panels mark places from which additional parametrization of convection lifts the CLaMS air parcels from the PBL into the upper troposphere.

To take into account the contribution of latent heat release to the vertical instabilities, $N^2$ can be modified by introducing the moist Brunt-Vaisala frequency $N_m$ with $N_m^2 = (g/\theta_e)(d\theta_e/dz)$ where $\theta_e$ is equivalent potential temperature, i.e. the temperature an air parcel would reach if all the water vapor in the air parcel were to condense, releasing its latent heat and then were brought down adiabatically to 1000 hPa (for details see also appendix A). The zonal mean of the dry and moist BVF distribution during boreal winter and summer 2005 is exemplary shown in Fig. 2.

Thus, lowest values of the dry BVF are distinctive for the middle troposphere especially in the tropics. The minimum in

5  lapse rate $d\theta/dz$ above the PBL represents the level of maximum convective impact (Gettelman and de Forster, 2002) which is also characterized by a vertically broad maximum of tropospheric signatures with enhanced CO and strongly reduced ozone (Pan et al. (2017) and references therein). Obviously, the values of $N^2$ clearly increase by crossing the tropopause but they also increase in the region below 700 hPa and are the lowest in the tropical and sub-tropical middle troposphere (i.e. within the

yellow contour in the top panel of Fig. 2). In the following section, we use this middle tropospheric minimum of $N^2$, although
zonally-resolved, to parametrize tropospheric mixing in CLaMS and, in this way, to include unresolved mixing processes in
the middle and upper troposphere into the CLaMS transport. Finally, we note that simple zonal means shown in Fig. 2 instead
of tropopause-based zonal means smear out the strong stability contrasts across the tropopause (Birner, 2006; Birner et al.,
2006). However, we use the climatology shown in Fig. 2 only to explain our procedure; the presented transport extension is
based on full 3d stability fields which contain all the details of the tropopause as far as they are resolved by the reanalysis.

On the other hand, low values of the moist BVF (i.e. bluish regions with $N_m^2 < 0$ in the bottom panel of Fig. 2) can be found
at altitudes below 700 hPa between 30°S and 30°N, i.e. in the tropical and subtropical PBL. The air parcels with $N_m^2 < 0$ are
conditionally unstable (see appendix A), i.e. they may undergo strong vertical uplift, if some other favorable condition causing
saturation may happen within such air masses (e.g. gravity wave-induced temperature fluctuations which are not resolved in the
meteorological data). In the following section, we show how conditional instability can be used to trigger additional (advective)
upward transport due to unresolved convection.

## 3 Extension of transport scheme: tropospheric mixing and unresolved convection

To extend the CLaMS mixing scheme, we follow two heuristic ideas: First, due to a much lower vertical stability in the
troposphere than in the stratosphere, we enhance tropospheric mixing in the model everywhere where (dry) vertical stability
is sufficiently small. Second, we take into account additional transport driven by convection, especially by deep convection
which is not sufficiently resolved in the reanalysis data.

Thus, whereas the first approach is related to changes in the mixing part of CLaMS and affects the next neighbors of each
Lagrangian air parcel, the second goal is related to changes in the advection part of CLaMS, i.e. to modification of the trajectory
calculation. Both extensions should be driven by instabilities quantified in terms of the dry and moist BVF, respectively, which
were introduced in the previous section. By including such a revised transport scheme, we seek for a better representation of
transport in the free troposphere, which also likely improves the performance of the model within the UTLS region. Because all
our changes are confined to the troposphere, we expect a weak influence on stratospheric transport in CLaMS which has been
successfully validated in many previous studies. Furthermore, the scheme should not give a heavy burden on the computation
time compared to the current version of CLaMS. Before going into the details, we shortly describe the standard version of
CLaMS (in the following denoted as the reference setup).

### 3.1 Reference setup

As the reference, we use the 2005-2008 timeslice of the 40-years CLaMS transient simulation started on 1st January 1979 and
driven by the horizontal winds and diabatic heating rates (vertical velocities) derived from the ERA-Interim reanalysis (Dee
et al., 2011; Ploeger et al., 2010). This configuration as well as the model initialization follow the model setup described in
(Pommrich et al., 2014) (100 km horizontal/400 m vertical resolution around 380 K, CLaMS v1.0). The first 10 years of the
CLaMS transient simulation can be considered as the model spinup. Zonal mean distribution of the mean age of air (AoA),

calculated relative to the Earth's surface is exemplarily shown for one day (19.08.2005) as the function of the latitude and the hybrid potential temperature $\zeta$ (Fig. 7a) and will be used for comparison with CLaMS simulations including the extended tropospheric transport (section 4).

The vertical coordinate $\zeta$ is the hybrid $\sigma$-$\theta$ which allows to resolve transport processes in the troposphere influenced by the orography and transport processes in the stratosphere where adiabatic horizontal transport dominates (Mahowald et al., 2002). More precisely, we replace $\eta$ by $\sigma$ in the hybrid $\eta$-$\theta$ coordinate as proposed by Mahowald et al. (2002), i.e.:

$$\zeta := f(\sigma)\theta(p,T), \tag{1}$$

with

$$f(\sigma) = \begin{cases} \sin\left(\frac{\pi}{2}\frac{1-\sigma}{1-\sigma_r}\right) & \sigma > \sigma_r \\ 1 & \sigma \le \sigma_r, \quad \sigma_r = \frac{p_r}{p_0} \end{cases} \tag{2}$$

Here, $p_0$ denotes the constant reference pressure level set to 1000 hPa. $p_r$ defines the pressure level around which the dry potential temperature $\theta$ smoothly transforms into the terrain-following coordinate $\sigma = p/p_s$.

For $p_r$ Mahowald et al. (2002) used the value 300 hPa, i.e. $\sigma_r = 0.3$. For situations with no orography (e.g. see surface with $p_s = 1000$ hPa) this means that the condition $\sigma = p/p_s < \sigma_r$ is valid everywhere above the level of 300 hPa. Consequently, in this region the vertical coordinate $\zeta$ is given by the dry potential temperature $\theta$. Conversely, below the level of 300 hPa, $\theta$ smoothly transforms into $\sigma$. For situations with orography (e.g. at the summit of Mount Everest with $p_s \approx 330$ hPa), the condition $\sigma = p/p_s < \sigma_r$ is first valid everywhere above $\approx 100$ hPa. Note that all $p_r$ values between 0 and 1000 hPa are possible with the consequence that higher $p_r$ values extend the applicability of $\theta$ as a vertical coordinate down to the lower troposphere. In the here used setup $\zeta$ covers the range between $\zeta = 0$ (i.e. orography-following) and $\zeta = \theta = 2500$ K (Pommrich et al., 2014).

Thus, because $\zeta$ in the troposphere is a less intuitive coordinate (hybrid mixture between $\sigma$ and $\theta$), isolines of pressure and potential temperature as well as the zonal mean of the WMO tropopause are also shown in Fig. 7a. Furthermore, vertical boundaries of the layers $\Delta\zeta_i$, $i = 1,\ldots,N$ within which CLaMS mixing is organized (for more details see Konopka et al. (2007)) are depicted at the right side of Fig. 7a. In the CLaMS reference run, the lowest layer $\Delta\zeta_0$ approximating the PBL extends between $\zeta = 0$ (Earth's surface) and 100 K, i.e. $\Delta\zeta_0 = \Delta\zeta_{pbl} = 100$K. After each trajectory step $\Delta t$, mixing ratios of all air parcels within this layer are replaced by their initial configuration and prescribed by a lower boundary condition (the same procedure is applied for the upper boundary $\Delta\zeta_N = 300$K covering the $\zeta$-range between 2200 and 2500 K.

In the default mixing scheme, CLaMS uses the integral deformations $\gamma = \lambda\Delta t$ derived from the relative motion of the next neighbors within each layer $\Delta\zeta_i$ (adaptive grid procedure). Here, $\lambda$ is the Lyapunov exponent of such a deformation and $\Delta t$ denotes the advective time step (typically between 6 and 24 hours). In the stratosphere and in large parts of the UTLS, where the flow is characterized by a high static stability, only sufficiently strong deformations with $\gamma > \gamma_c$ ($\gamma_c$ denoting an empirical critical deformation) are expected to trigger mixing with the best choice for $\gamma_c$ between 0.8 and 1.5 (Konopka et al., 2004). This also means that there is some freedom in the choice of the parameters $\lambda$ and $\Delta t$. Whereas for the stratosphere the values of $\Delta t = 24$ h and $\lambda = 1.5$ day$^{-1}$ ($\gamma_c = 1.5$) were used in the past (Konopka et al., 2004) (and are used here as the reference

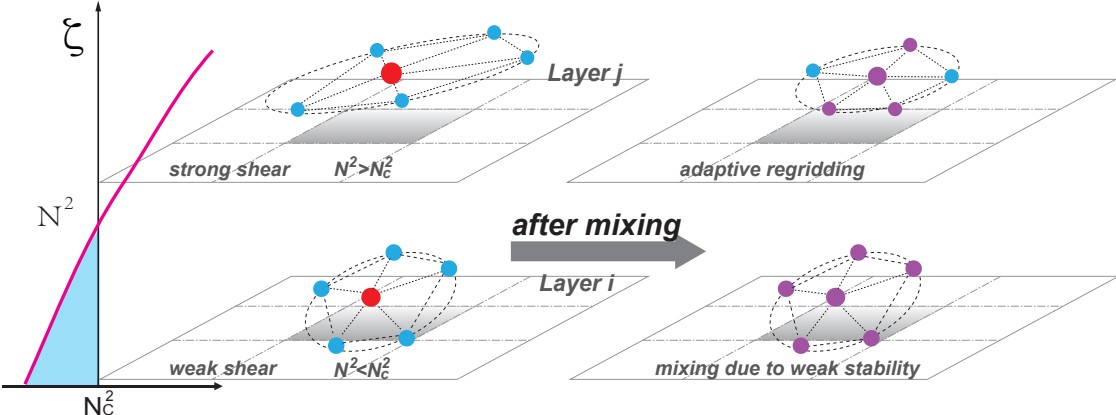

**Figure 3.** Mixing driven by vertical instability and strong wind shear. The profile on the left side is an idealized $N^2$ profile derived from the reanalysis and interpolated on the CLaMS air parcels (red). The lower part is representative for the troposphere with $N^2 < N_c^2$ and the upper part for the stratosphere with $N^2 > N_c^2$. Thus, in the lower layer static stability is weak with the opposite configuration in the upper layer where, in addition, the wind shear is strong. In the upper layer, adaptive regridding is used to include mixing between a subset of the next neighbors of the considered air parcel (default mixing in CLaMS). In addition, in the lower layer, all next neighbors will be mixed with the considered air parcel if criterion $N^2 < N_c^2$ is valid. The purple parcels are mixed parcels of red and blue parcels.

configuration), a larger mixing frequency with $\Delta t = 6$ h seems to give better results in the well-mixed troposphere and in the UTLS region (Vogel et al., 2011; Konopka and Pan, 2012).

## 3.2 Tropospheric mixing

In the following, we assume that the additional tropospheric mixing should be triggered whenever the corresponding value of $N^2$ interpolated at the CLaMS air parcel is less than a critical value denoted in the following as $N_c^2$. $N_c^2$ is a free parameter

5   which, basically, can be adjusted by comparison with the experimental data. We expect that $N_c^2$ should be around zero and should identify regions with (statistically) enhanced tropospheric mixing. If an air parcel fulfills the criterion $N^2 < N_c^2$, the air parcel will be mixed with all next neighbors diagnosed by Delaunay triangulation in the respective CLaMS layer under consideration (so we use the same next neighbors as in the standard CLaMS mixing scheme). In this way the composition of the considered air parcels are affected, but not their geometric positions.

10   Figure 3 is a schematic diagram illustrating how the tropospheric extension of mixing works. If this additional mixing is applied, we set the new mixing ratio of a considered air parcel and its next neighbors through averaging their composition, which is shown as a change of the parcel's color. This setting completes the mixing without changing any model parcel position and the number of parcels because it can be executed directly in the current mixing module (more precisely, after deformation driven adaptive grid procedure of CLaMS).

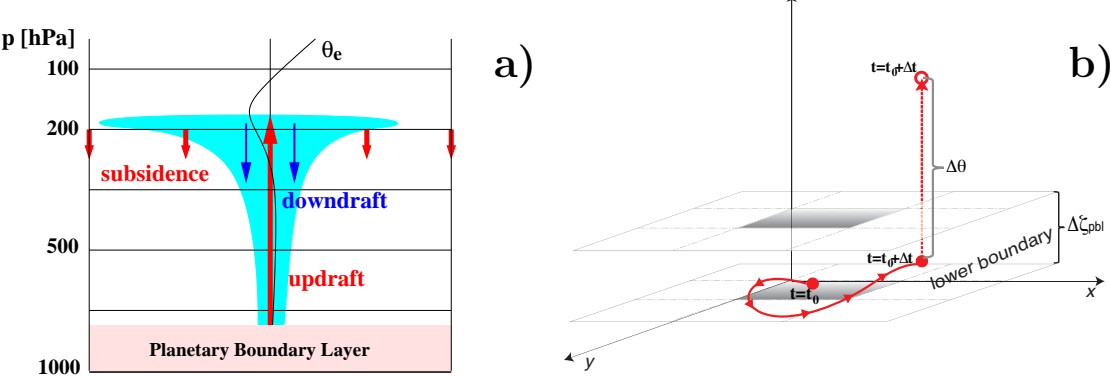

**Figure 4.** a) Schematic definition of convective fluxes and scales. Vertically unstable profile of the equivalent potential temperature $\theta_e$ (black thin line) triggers convective updraft in our simplified scheme. b) Convective updraft in CLaMS: the original trajectory (thick solid red) and the modified trajectory (dashed red) with convective transport from the lowest CLaMS layer $\Delta\zeta_{pbl}$ at time step $t_0 + \Delta t$. The vertical displacement $\Delta\theta$ of the considered air parcel is estimated through the latent heat release of condensation and may happen at every time step along the trajectory when the reanalysis data is read (i.e. every $\Delta t = 6$ hours for ERA-Interim, see text for more details).

### 3.3 Unresolved (deep) convection

Commonly, convection is understood as a vigorous vertical updraft resulting from instabilities in the vertical temperature and water distribution profiles, preferably connecting the PBL with the free troposphere and is schematically shown in Fig. (4a). Convection starts on relatively small horizontal scales of the order of few kilometers and is mainly driven by the latent heat
release of the gaseous and liquid water content. Such diabatic processes may lift a large amount of air. The respective upward mass flux is traditionally denoted as an updraft (thick upward red arrow in Fig. 4a).

The reverse diabatic mass transport, the so-called downdraft is related to the evaporation of water or melting of ice and is in general much smaller than the original convective updraft (thin blue arrows). A much larger part of the downward mass flux, the so-called subsidence, occurs on much larger horizontal scales of the order 10-100 km, or even larger if Rossby- or
gravity-waves are induced by convection (thick downward red arrows). A significant part of subsidence results mainly from long-wave cooling of radiatively active water vapor which is vertically transported and freeze-dried within convective tower and subsequently spread horizontally at the level of main convective outflow. Note that the horizontal scale of this outflow region is at least one order of magnitude larger than the horizontal extension of the convective tower at the ground.

It is generally believed that the exchange of mass driven by deep convection can efficiently inject the air masses from the PBL
into the upper troposphere or even, although very rarely, into the lower stratosphere (Schiller et al., 2009; Corti et al., 2006). In fact, the extension of mixing presented in the previous subsection is still limited by the model layers and, consequently, is not suitable to parametrize unresolved convective events which connect the PBL with layers in the upper troposphere on a time scale of minutes to hours.

Now, we present an alternative method to enhance upward transport for conditionally unstable air parcels with $N_m^2 < 0$ in order to lift such air masses from the lowest layer of CLaMS $\Delta\zeta_{pbl}$ (following the orography and approximating here the PBL) into the upper troposphere. Figure 4b shows the concept of estimating the uplift of boundary air by adding a $\Delta\theta$ to the trajectory in the vertical direction when the condition $N_m^2 < 0$ is diagnosed along the trajectory (i.e. using diabatic concept of vertical motion).

Following Ertel (1938) (for details see appendix B), we use the following approximation for $\Delta\theta$:

$$\Delta\theta = \frac{L_v \theta_0 \mu_w}{c_p T}, \tag{3}$$

where $\theta_0$ and $\mu_w$ denote the potential temperature and the total water vapor mass mixing ratio in the air parcel where the condition $N_m^2 < 0$ is fulfilled. $T$ denotes the temperature in the air parcel and approximates here the saturation temperature. $L_v$ is the specific latent heat for evaporation and $c_p$ denotes the specific heat at constant $p$.

To illustrate, how such a parametrization works, the zonally-resolved fraction of events with $N_m^2 < 0$ occurring within the lowest CLaMS layer are shown in Fig. 5. The respective DJF and JJA climatologies derived from ERA-Interim for 2005 reveal the expected spatial distribution, land-ocean contrasts as well as the seasonality. To justify the use of $N_m^2 < 0$ as a proxy of convection, we compare its spatial distribution with the satellite-based Outgoing Longwave Radiation (OLR, top row, cyan isolines) as well as with the ERA-Interim-based convective available potential energy (CAPE) as an alternative

method to detect convection (for definition of CAPE, see appendix B). The comparison shows a good correlation between the climatology of the regions with $N_m^2 < 0$ and the respective OLR (top panel of Fig. 5) whereas the correlation with the CAPE is less pronounced (e.g. in the region around 30-50E, 45°N during JJA). Motivated by this finding and because of the simplicity of tracing conditionally unstable air parcels, we use in the following the criterion $N_m^2 < 0$ as the first condition to trigger convective events.

Our second condition is related to the question if every conditionally unstable air parcel is a source of convection which should be taken into account. This question is also related to the fact that the number of air parcels in CLaMS is not strictly

conserved but kept roughly constant within about $\pm 10\%$ flexibility through the adaptive regridding procedure (current mixing scheme). It means that the mixing procedure is able to adjust a certain increase or decrease in the number of air parcels, but this amount should be below $\pm 10\%$. Figure 6 shows the one-year climatological PDF of $\Delta\theta$ in the tropics (30°S- 30°N). The probability of $\Delta\theta$ larger than 35 K is around 30% and it decreases rapidly from 35 K to 60 K. When the $\Delta\theta$ is too small to leave the lower boundary, it is not necessary to add $\Delta\theta$ to the trajectory. Thus, as our second condition, we only uplift such air

parcel along the trajectory (i.e. within the advective step) if $\Delta\theta$ is sufficiently large. In our control runs, the minimum of $\Delta\theta$ is set to $\Delta\theta_{crit} = 35$ K and is shown as the dashed line in Fig. 6. This choice is also the reason that we call this parametrization "deep" convection scheme.

To estimate the influence of our convection parametrization on the mass budget, we calculate the annually and globally averaged mass flux due to paramtrized convective updrafts (for details see appendix C). Because only updrafts are parametrized,

this has an impact on the total mass budget in CLaMS with a surplus of the annually and globally averaged mass flux density of around 10 kg/m$^2$day between the upper edge of the lowest CLaMS layer and around 360 K (highest level of the here

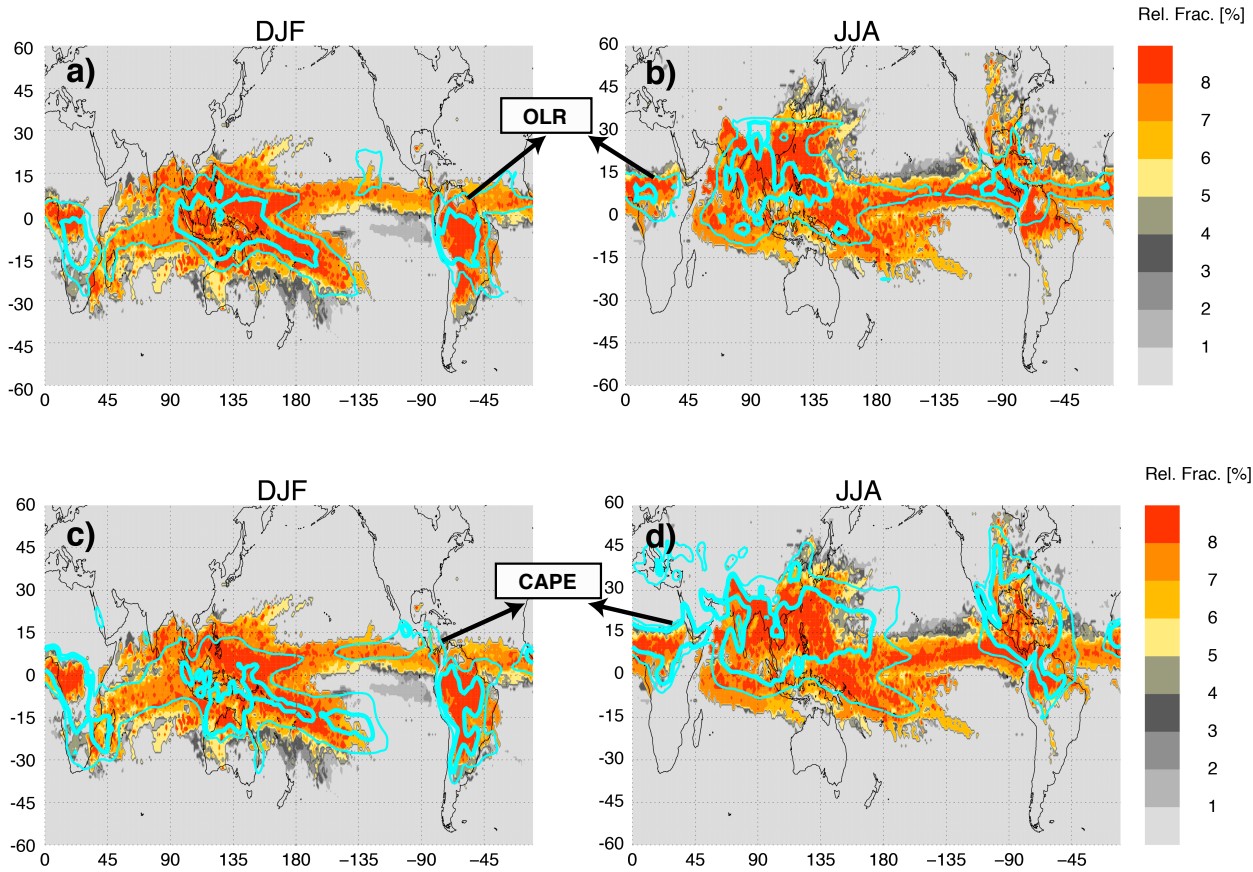

**Figure 5.** Color-coded is the fraction of ERA-Interim time steps (6h-frequency) within a season when the criterion triggering the deep convection scheme is fulfilled at CLaMS air parcels within the lowest layer of the model $\Delta\zeta_{pbl} = 250$K ($\sigma_r = 0.7$, for details see text). Left and right column show DJF and JJA 2005 climatologies, respectively. In the top row the isolines of the Outgoing Longwave Radiation (OLR) as derived from the NOAA satellite archive are highlighted while in the bottom row the isolines of the convective available potential energy (CAPE) are overplotted (cyan).

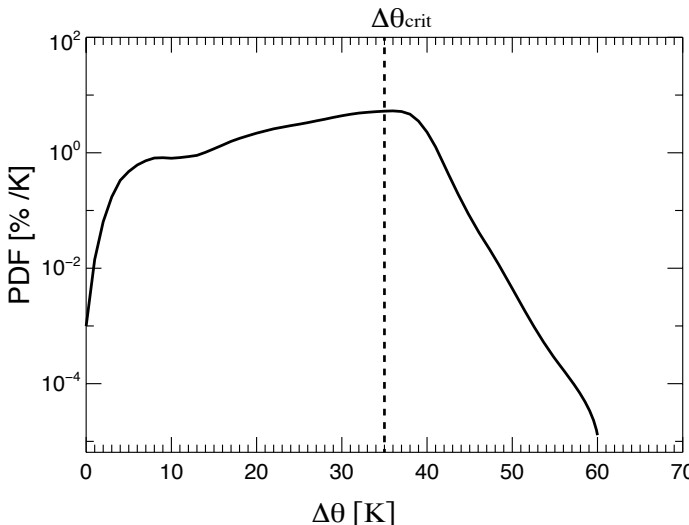

**Figure 6.** The probability distribution function (PDF, in units of %/K) of convective uplift $\Delta\theta$ in the tropics (30°S-30°N) for the air parcels in the lowest level of CLaMS ($\Delta\zeta = 250$ K, $\Delta z \approx 1.4 - 2$ km) where $N_m^2$ is negative using the ERA-interim reanalysis data of the year 2005. The restriction used for the "deep" convection scheme ($\Delta\theta > \Delta\theta_{crit} = 35$ K) is marked as the dashed straight line.

parametrized convection). On the other hand, the used ERA-Interim diabatic vertical velocities are *per se* not mass-conserving. Fueglistaler et al. (2009) have shown that there is a significant deficit of mass of the order of 20 kg/m$^2$day (c.f. their Figure 10), roughly in the same altitude range where the CLaMS convective parametrization works.

A possible explanation for this deficit can be related to the fact that the horizontal resolution of the ERA-Interim reanalysis (roughly 80 km) does not sufficiently resolve the convective towers (which are of the order 1 km) but does better resolve the large-scale subsidence which may be better reproduced in the ERA-Interim diabatic vertical velocities. Thus, our parametrization aims to close this in-balance by including some additional convective updrafts. The results show that qualitatively our simple approach roughly balances such a deficit and there is still some potential to enhance the strength of parametrized con-

vection. At least the influence of our parametrization on the mass budged is comparable with "intrinsic uncertainties" in the mass budget of the reanalysis itself. Following the procedure described in Rosenlof (1995) and applied in Konopka et al. (2010) (see their appendix), it is possible to restore the mass balance violations at least in the annual mean. It is also worth to not that unresolved convection is a process not present in the divergence of the horizontal wind and, consequently, not influencing the 3d (large-scale) velocity field.

Finally, it should be mentioned that our Lagrangian parcels are still too large (around 100 km in horizontal and few hundred meter in vertical direction) to be transported by realistic convective systems. Thus, they are more suitable to describe large-scale convective outflow rather than the convective towers which are well below their horizontal resolution. This fact justifies to a certain extent our restriction to consider only "deep" convective events with $\Delta\theta > 35$ K. Note also that by increasing model

**Table 1.** List of CLaMS reference and control simulations with different configurations of mixing, unresolved convective updrafts and parameters $\sigma_r$ defining the transition between the $\sigma$- and $\theta$-related vertical coordinate. For all simulations with unresolved convective updrafts and tropospheric mixing, the critical values of $N_m^2$ and $N^2$ are set to 0 and $1 \cdot 10^{-4}$ s$^{-2}$, respectively, and $\Delta\theta_{crit} = 35$ K.

| Name | time step $\Delta t$ [hr] | $\lambda_c$ [day$^{-1}$] | unres. convection | trop. mixing | $\sigma_r$ | $\Delta\zeta_{pbl}$ [K] |
|---|---|---|---|---|---|---|
| REF | 24 | 1.5 | N | N | 0.3 | 100 |
| REF_0.7 | 24 | 1.5 | N | N | 0.7 | 250 |
| REF_0.3 | 24 | 1.5 | N | N | 0.3 | 140 |
| MIX_0.7 | 6 | 3.5 | N | N | 0.7 | 250 |
| TROP_MIX_0.7 | 6 | 3.5 | N | Y | 0.7 | 250 |
| UNRES_CONV_0.7 | 6 | 3.5 | Y | N | 0.7 | 250 |
| FULL_EXT_0.7 | 6 | 3.5 | Y | Y | 0.7 | 250 |
| FULL_EXT_0.3 | 6 | 3.5 | Y | Y | 0.3 | 140 |
| FULL_EXT_LM_0.7 | 6 | 4.0 | Y | Y | 0.7 | 250 |

resolution by a factor of 2, both horizontally and vertically, the respective convective mass flux would decrease by a factor of 8.

## 4 CLaMS performance with additional tropospheric transport

In this section, we describe the details of the CLaMS configuration for the simulations with extended tropospheric transport (v2.0), show in which part of the atmosphere the CLaMS air parcels are affected by this extension and compare the respective AoA distributions. Especially, we investigate the propagation of the $CO_2$ distribution from the boundary layer into the lower stratosphere for different model configurations and evaluate the related $CO_2$ variability (annual cycle and trend) with the observations and the CarbonTracker model simulations.

### 4.1 Setup for reference/control simulations

Like for the reference simulation described in the subsection 3.1 (in the following denoted as REF), all other simulations cover the same time period 2005-2008 and have the same vertical and horizontal resolution above the tropopause. All these simulations start on 01.01.2005 and are initialized with the REF concentrations. The length of 3 years for all our simulation is long enough to see the changes in the troposphere and tropical lower stratosphere. However, 3 years are certainly too short for the whole stratosphere but this is our compromise concerning the available computing time and is long enough to see at least in which direction the changes of the model have to be expected. Table 1 provides the key information for all CLaMS simulations discussed in this paper.

In addition to the REF configuration, we use two slightly different configurations: $\sigma_r = 0.7$, $\Delta\zeta_{pbl} = 250$K (REF_0.7, Fig. 7b) and $\sigma_r = 0.3$, $\Delta\zeta_{pbl} = 140$K (REF_0.3, Fig. 7c) with the letter choice being very close to the REF case with $\sigma_r = 0.3$

and $\Delta\zeta_{pbl} = 100K$ (see below). Grid configurations for all cases listed in Table 1 are the same in the stratosphere although significant differences are in the troposphere: By using $\sigma_r = 0.7$, isentropic mixing and diabatic vertical velocities, two central concepts of CLaMS, can be extended to a larger part of the troposphere, in particular to the middle tropical troposphere. Note that the 320, 330K isentropes in Fig. 7b are within CLaMS layers defined by the $\zeta$ coordinate and, consequently, mixing within such layers is roughly isentropic. Thus, almost the whole UTLS region, down to the tropical middle troposphere, is covered by such isentropic layers in the $\sigma_r = 0.7$ simulation (see Tao et al. (2018), especially their appendix 1).

Furthermore, both apparently different choices of $\Delta\zeta_{pbl}$ for REF_0.7 and REF_0.3 correspond roughly to the same geometric thickness of the lowest CLaMS layer, which varies between 1.4 and 2.2 km and approximates here the PBL. Because in the REF case this thickness is certainly too small (between 1.1 and 1.3 km), we decided to enhance these values to more realistic numbers. Thus, for a more fair comparison, we take REF_0.7 or REF_0.3 as the main reference (instead of REF) with $\sigma_r = 0.7$ being our main choice. It should be emphasized that by using predefined boundary conditions in the PBL, we do not resolve any transport in this part of the atmosphere and confine our efforts only to improve transport in the free troposphere extending between the PBL and the tropopause.

As a first step toward extended tropospheric transport (in the following denoted as control runs), we decrease the advective time step of trajectories from 24 to 6 hours and, to keep the intensity of the standard CLaMS mixing scheme roughly constant, we also increase the Lyapunov exponent from 1.5 to 3.5 day$^{-1}$ corresponding to the critical deformation $\gamma_c = 0.85$ (case MIX_0.7 in Table 1). Such a slightly higher mixing frequency relative to the reference case (every 6 instead of every 24 hours) has proven to give better representation of the CO-ozone correlations in the UTLS region (Vogel et al., 2011; Konopka and Pan, 2012). Higher mixing frequency resolves also the diurnal cycle of our new parametrization of tropospheric transport described in the last section.

The next step is to add new tropospheric transport, i.e. tropospheric mixing (TROP_MIX) and unresolved convective updrafts (UNRES_CONV). Before discussing this new contributions we exemplary show our results in Figure 7. Here, the zonal mean distribution of AoA, calculated relative to the Earth's surface for one day (19.08.2005) is plotted for the three cases discussed above: REF (Fig. 7a) and two control simulations with full extension of tropospheric transport FULL_EXT_0.7 (Fig. 7b) and FULL_EXT_0.3 (Fig. 7c). Both types of control simulations show much younger air in the troposphere if compared with the reference run. Also the gradients across the tropopause are more pronounced. On the other hand, stratospheric distributions are very similar for all three cases. Now, we go more into the details of such a tropospheric extension.

## 4.2  Diagnostic of extended tropospheric transport

It is easy to tag and count all air parcels in CLaMS which undergo additional tropospheric mixing and which are lifted from the lower boundary to the middle and upper troposphere by the deep-convection scheme introduced in the previous section. In Fig. 8a/b the DJF/JJA zonally averaged fractions of additionally mixed air parcels (calculated for 2005) are color coded as the function of latitude and pressure. The black isolines in the top panel approximate the fraction of the CLaMS air parcels which were lifted from the lowest boundary layer to the middle and upper tropical troposphere using the deep convection parametrization. For comparison, the fractions of the number of air parcels affected by the default CLaMS mixing scheme

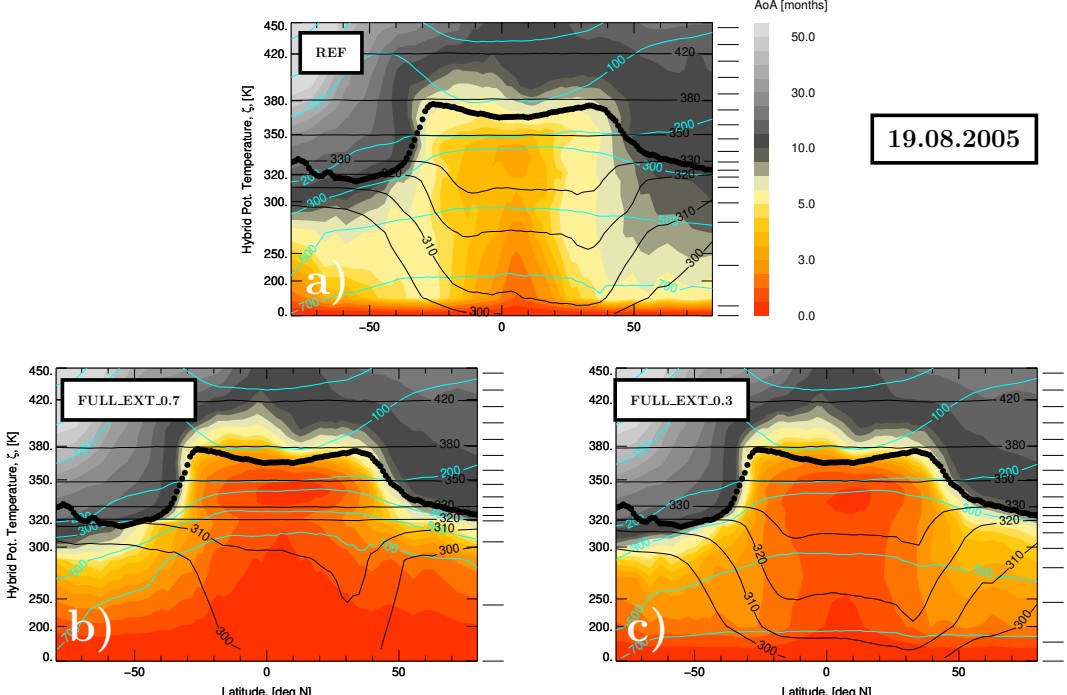

**Figure 7.** Zonal mean of mean age (AoA) for the reference simulation (a) and for the simulations with the full extension of tropospheric transport (bottom) exemplarily calculated for 19.08.2005 (i.e. after more than 8 months of transport). For the definition of the abbreviations see Table 1. The hybrid vertical coordinate $\zeta$ is used and the isolines of the potential temperature $\theta$ (black), pressure $p$ (cyan) as well as the zonal mean of the WMO tropopause are also shown. On the right side of each panel, the boundaries of the CLaMS layers $\Delta\zeta_i$, $i = 1, \ldots, N$ are plotted (see text for more details).

are shown in the bottom panels of Fig. 8, which always happens, independent if the here discussed extension of transport is included or not. In addition, the respective mean WMO tropopause as well as the isentropes are shown.

Note that CLaMS tropospheric mixing practically does not affect any air parcels in the stratosphere (the zero line of the calculated fraction is well below the tropopause, not shown). Note also that numbers of air parcels affected by the deep convection scheme are smaller than 10% with highest levels (around 360 K) during JJA, mainly related to the Asian Summer Monsoon (not shown). Furthermore, both tropospheric mixing and the deep convection transport show some seasonality like $N^2$ and $N_m^2$, respectively (i.e. higher in the summer hemisphere). Finally, the default mixing scheme is much weaker than

the tropospheric mixing although the seasonality is very similar. Note that this part of mixing is also present in the vicinity of the tropopause although with stronger signatures on the tropical side of the jets and below the tropopause. At least in this climatological picture, only standard mixing in CLaMS contributes to a direct STE.

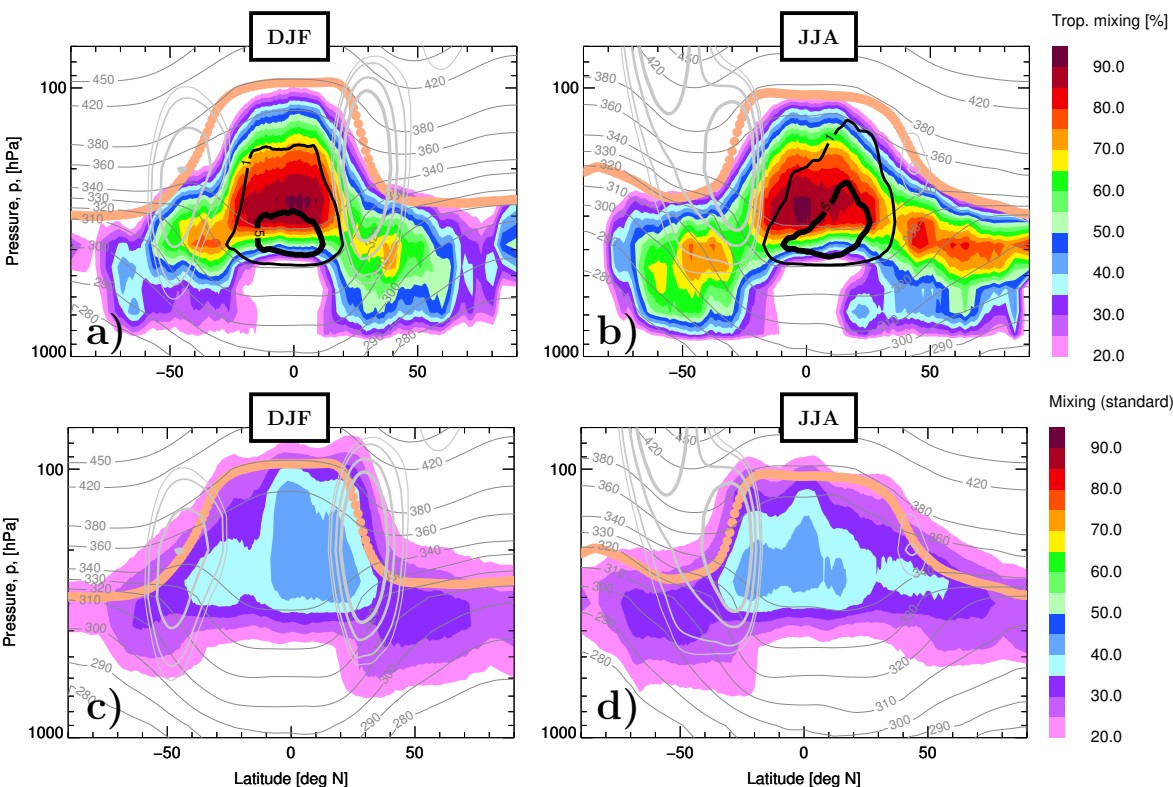

**Figure 8.** Top: DJF/JJA 1-year (2005) climatology of percentage of CLaMS air parcels undergoing tropospheric mixing (colors) and of air parcels lifted from the lowest layer of the model into the middle and upper (tropical) troposphere (black contours). Bottom: Same type of climatology but for air parcels which undergo the standard CLaMS mixing procedure (adaptive regridding driven by horizontal strain and vertical shear within the CLaMS layers). The respective WMO tropopause (beige), horizontal wind marking the position of the jets (light gray) and the isentropes (dark gray) are also shown.

## 4.3 Validation with $CO_2$ observations

$CO_2$ is a useful tracer for validation of transport in the models, mainly in the troposphere and lower stratosphere where
$CO_2$ is basically chemically inert (Waugh and Hall, 2002). The only stratospheric source of $CO_2$ is a small contribution (<1 ppmv) from methane oxidation (Bates and Nicolet, 1950; Ehhalt, 1974; Boucher et al., 2009) that is taken into account in all CLaMS simulations discussed here (Pommrich et al., 2014). Thus, the quality of the $CO_2$ distribution reproduced in CLaMS is determined largely by the quality of the lower boundary condition and the quality of the representation of transport. For the latter, tropospheric transport, is a significant part.

The atmospheric mixing ratios of $CO_2$ are essentially both monotonically increasing (trend) and periodic (seasonality) signals which define a stringent test for the model representation of tropospheric transport and STE (Bönisch et al., 2008, 2009). As recently shown by Diallo et al. (2017), even inverted vertical profiles of $CO_2$ across the extratropical tropopause

are possible during the Northern Hemispheric summer despite of a continuous increase of the mean $CO_2$ in the PBL resulting from the growing anthropogenic emissions.

In CLaMS, $CO_2$ mixing ratios propagate upwards from the lowest layer $\Delta\zeta_{pbl}$ for which the CarbonTracker data set was used (Peters et al. (2007), the updates are documented at http://carbontracker.noaa.gov) with $CO_2$ mixing ratios available for the 2000-12 period (simulation CT2013B, available every 3 hours, see ftp:/products/carbontracker/co2/CT2013B/molefractions/co2_total/).

In particular, the first five lowest levels of each CarbonTracker data set were vertically averaged and used to overwrite CLaMS air parcels within the PBL layer every 6 hours. The reference simulation was initialized at 01.01.2000 and beginning from 01.01.2005 all other control simulations were started using the output of the reference simulation for the initial distribution.

The zonal means of $CO_2$, exemplarily calculated for two representative days, 5th of Mai and 25th of September, 2005, are shown in Fig. 9. In particular, results for the reference simulation (REF) and for the two control simulations with full

tropospheric transport (FULL_EXT, $\sigma_r = 0.3/0.7$) can be compared with the respective CarbonTracker distribution which was used in CLaMS to initialize the lower boundary of the model. In all $CO_2$ distributions, the upward propagation of the annual cycle can be clearly diagnosed with higher values during the boreal summer and vertical inversion during the fall. However, the propagation of the tropospheric signal shows some obvious differences with a faster upward propagation in CLaMS control simulations than in the CLaMS reference run. Note that the cross-hemispheric transport is weaker in the

$\sigma_r = 0.7$ than in the $\sigma_r = 0.3$ CLaMS configuration. Note also that the upward propagation of the $CO_2$ annual cycle is well-confined by the position of the tropopause (black dots) in all CLaMS runs while in the CarbonTracker data this property is less pronounced, although like in CLaMS, ERA-Interim reanalysis is in the underlying transport model (see CT2013B documentation: https://www.esrl.noaa.gov/gmd/ccgg/carbontracker/CT2013B). We will come to this point later.

Now, the $CO_2$ time-space evolution derived from CLaMS simulations are compared with the observations of the Compre-

hensive Observation Network for TRace gases by AIrLiner (CONTRAIL) (Machida et al., 2008). $CO_2$ mixing ratios were measured during regular flights by Japan Airlines from Japan to Australia, Europe, North America, and Asia with continuous measuring equipment (CME) for in situ $CO_2$ observations, as well as improved automatic air sampling equipment (ASE) for flask sampling (for more details about the instrument see Machida et al. (2002)). This data set provides significant spatial coverage, particularly in the Northern Hemisphere (Sawa et al., 2015). CONTRAIL observations have a vertical resolution of a

few meters (during ascents and descents) and a horizontal resolution of a few hundred meters, resulting from the high sampling frequency of these instruments.

Here, we use the zonally and monthly averaged time evolution of these observations between 2005 and 2008 interpolated at a latitude-altitude grid with $10° \times 1$ km resolution and extending between 20°S to 60°N and 5.5-12.5 km, respectively (for more details see Diallo et al. (2017)). Comparison of these mean CONTRAIL observations with the respective CLaMS results

for the reference and all control simulations are shown in Fig. 10.

In particular, the comparison of the seasonal cycle and trend at 15°N for two selected altitudes of 5.5 and 10.5 km is plotted in the left panel of Fig. 10. The right panel shows the accumulated error, i.e. the zonal mean of the mean square deviation between the CLaMS simulation and CONTRAIL observations averaged over all latitude-altitude grid points where gridded (mean) measurements and their standard deviations are available. Note a higher variability of $CO_2$ at 5.5 than 10.5 km level

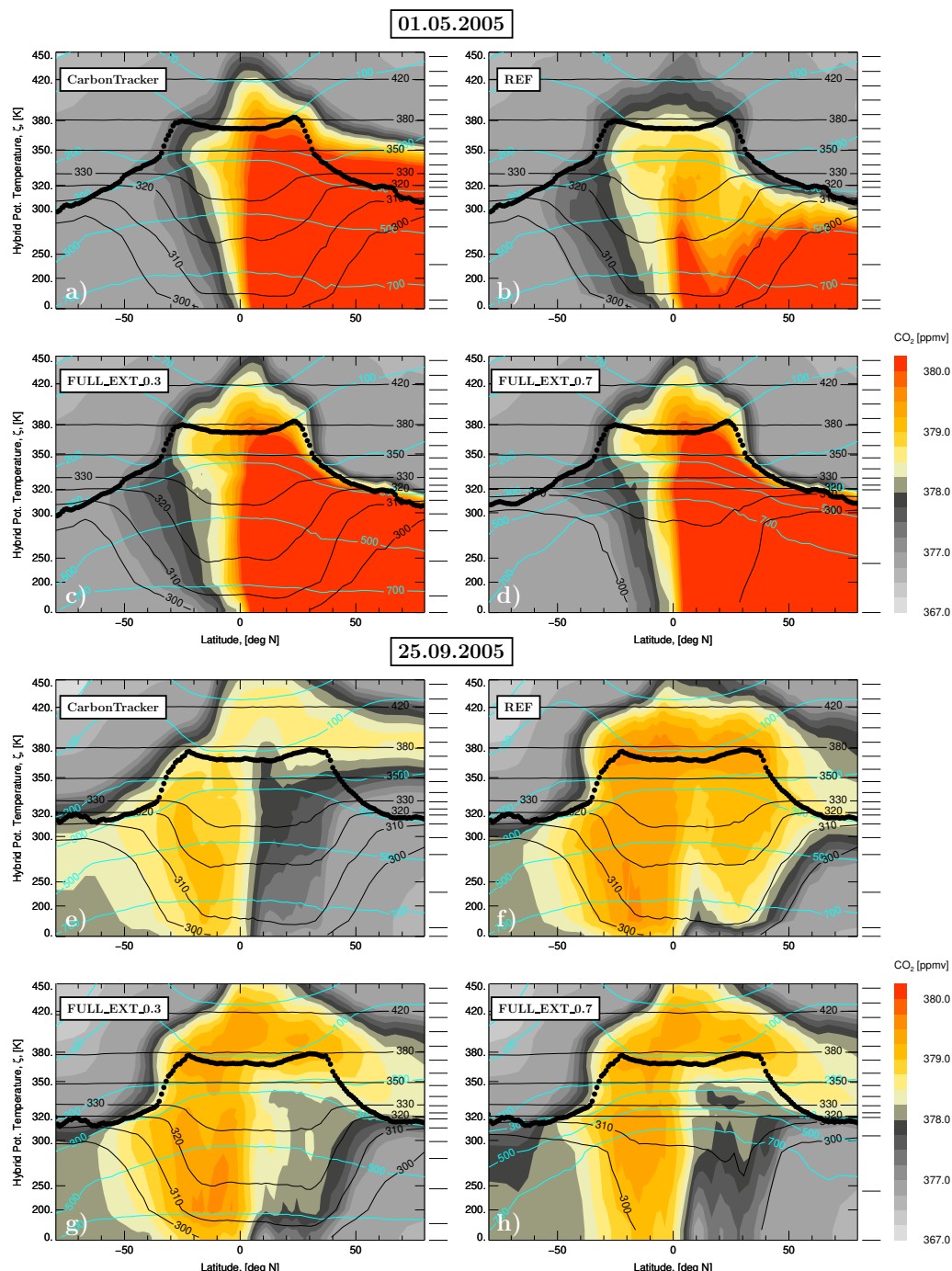

**Figure 9.** Upward propagation of the $CO_2$ distribution from the lowest layer of the model where $CO_2$ was initialized by the CarbonTracker data (CT2013B) on two exemplary days: 01.05.2005 (a to d) and 25.09.2005 (e to h). The CLaMS zonal means for different model configurations are compared with the CarbonTracker distribution itself. Black dots denote the tropopause derived from the ERA-Interim data.

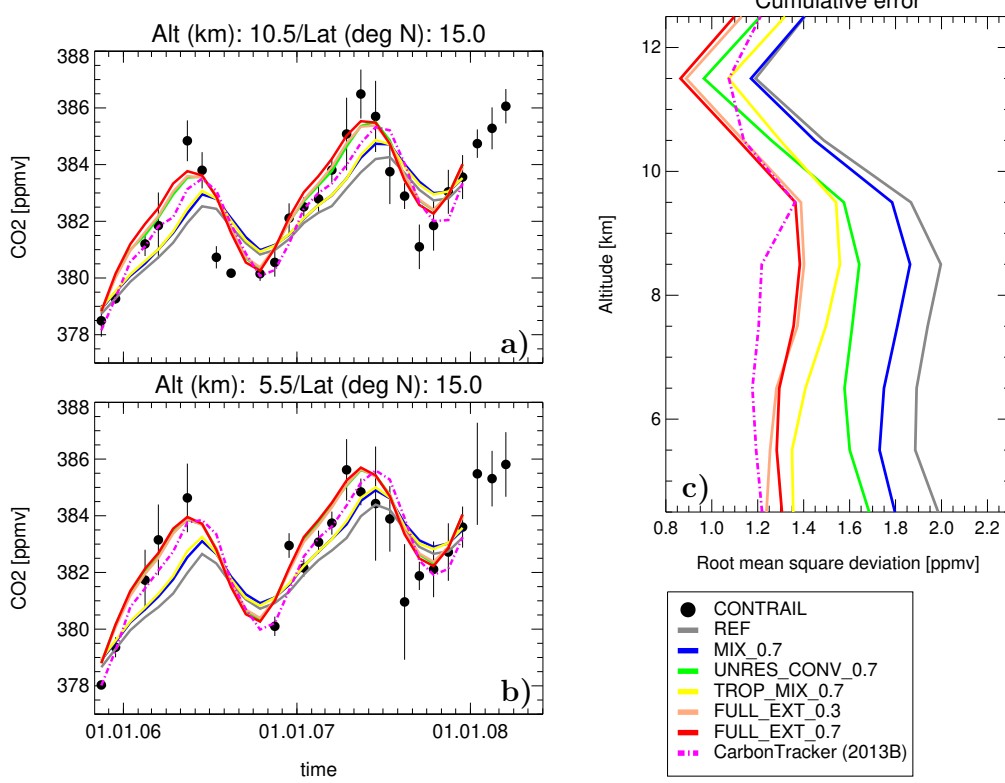

**Figure 10.** Left: Seasonality of $CO_2$ for CLaMS simulations compared with CONTRAIL observations (black filled circles/vertical lines denote mean values and their standard deviations, respectively) at two locations: $15°N$, 5.5 km (a) and $15°N$, 10.5 km (b) Right: The accumulated errors of $CO_2$ for CLaMS simulations compared with all CONTRAIL observations as the function of the altitude.

indicating a stronger variability due to the differences in the sources of $CO_2$ than due to vicinity of the tropopause where our gridding procedure does not differ between the stratospheric and tropospheric values. Thus, the reference simulation (REF) and the simulation with 6h mixing frequency (MIX_0.7) show not only too small amplitude but also their phase is delayed if
5  compared with the CONTRAIL observations.

There is a clear improvement of the representation of the $CO_2$ distribution quantified in terms of the phase and the amplitude of the seasonal cycle as well as in terms of the accumulated error by taking additional tropospheric transport into account. The best results are achieved by including both the tropospheric mixing and the convection parametrization (orange and red curves are for FULL_EXT_0.7 and FULL_EXT_0.3). By switching off the tropospheric mixing or the convection parametrization or
10  both (here the results only for $\sigma_r = 0.7$ are shown), the cumulative error increases up to 80%. While the tropospheric mixing is more important below 9 km, the improvement due to convection parametrization dominates between 9 and 13 km. Although there are still some errors in the amplitude, the additional tropospheric transport significantly improves the overall agreement. Note that also the CarbonTracker distribution, even achieved by assimilating observations does not show a perfect comparison

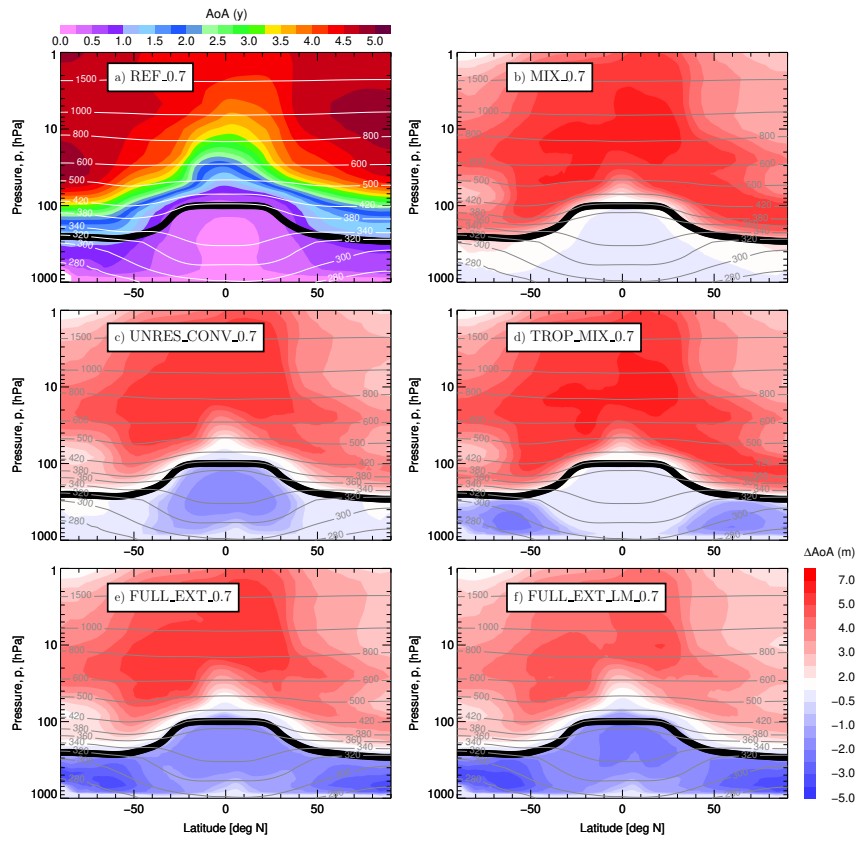

**Figure 11.** Annual and zonal mean of AoA (for 2007, in years) as derived from the reference simulation REF_0.7 (a) and the differences relative to this distributions (in month) calculated for some configurations listed in Table 1. Black line denotes the tropopause.

with the CONTRAIL observations (which are not included into the assimilation procedure of the here used version CT2013B). Remarkably, CLaMS control simulations are becoming even better than CarbonTracker distributions in the region above 10

5   km (c.f. Fig. 9) probably caused by a very limited vertical resolution of the CarbonTracker data around the tropopause (only 6 levels between 9 and 18 km).

### 4.4   Impacts on the stratosphere

For this purpose, we discuss in Fig. 11 the differences in the distribution of AoA due to extension of tropospheric transport by considering its annual and zonal mean calculated for the year 2007 (last year of our simulations covering the 2005-07 period). As a reference distribution we use the AoA of the REF_0.7 case shown in the Fig. 11a) and calculate the respective differences in the AoA distributions of the simulations with 6h default mixing (MIX_0.7, panel b), with added unresolved convective updrafts (UNRES_CONV_0.7, panel c), with added tropospheric mixing (TROP_MIX_0.7, panel d), with the full

tropospheric extension (FULL_EXT_0.7, panel e) and with full tropospheric extension but slightly reduced default mixing by
increasing $\lambda_c$ from 3.5 to 4.0 day$^{-1}$ (FULL_EXT_SM_0.7, panel f).

As expected, the air below the tropical tropopause becomes younger by up to 6 months if the additional tropospheric transport is included (panels (c) to (f)). The effect of convection is mainly confined to the tropics while the strongest impact of tropospheric mixing can be diagnosed in the high latitudes. However, and at first surprisingly, the air becomes slightly older in the stratosphere (around 6 months). Note that this is not a consequence of enhanced tropospheric transport but of the change
in the default mixing scheme from $\Delta t = 24$ hours, $\lambda_c = 1.5$ day$^{-1}$ (reference) to $\Delta t = 6$ hours, $\lambda_c = 3.5$ day$^{-1}$ and can be deduced from the panel (b). Such a change leads to a slightly enhanced isentropic mixing across the tropical pipe, which also enhances the stratospheric re-circulation and makes the stratospheric air older due to aging by mixing (Garny et al., 2014; Poshyvailo et al., 2018). Consistently, aging by mixing in the stratosphere becomes smaller by reducing the isentropic part of CLaMS mixing by setting $\Delta t = 6$ hours, $\lambda_c = 4.0$ day$^{-1}$ in the default mixing scheme (panel (f)) although the tropospheric
AoA is only weakly affected. This indicates a secondary role of CLaMS standard mixing scheme in the troposphere and underlines its primary role in the stratosphere. In this way, the cross-tropopause gradient of AoA can be enhanced by combining the stratospheric mixing with the proposed extension of tropospheric transport.

## 5   Conclusions

Implementation of mixing in Lagrangian transport models is still an important issue in the ongoing scientific discussion. Here,
we follow the idea of using numerical diffusion to parametrize physical mixing which was first proposed and implemented in connection with the Chemical Lagrangian Model of the Stratosphere (CLaMS). In particular, we extend this idea to the troposphere where vertical stability is much smaller if compared with the stratosphere for which CLaMS was originally developed. By using the lapse rates of the dry and moist potential temperature mainly defining the squares of the dry and moist Brunt-Vaisaila frequencies $N^2$ and $N_m^2$, we parametrize two important troposheric processes which are not sufficiently resolved in the
current version of CLaMS v1.0 (Pommrich et al., 2014), i.e.: tropospheric mixing in regions with small lapse rates of the dry potential temperature and unresolved (deep) convection in regions with conditionally unstable lapse rates of the moist potential temperature.

The implementation of both processes improves CLaMS performance measured here in terms of the quality of the simulated $CO_2$ seasonality. However, there is still some freedom in the choice of the free parameters (like critical values for $N^2$, $N_m^2$ or $\Delta\theta$ ) and in the representation of the PBL in the model. In particular the choice of the critical displacement $\Delta\theta$ triggering the onset of convective updrafts (currently set to 35K) needs further investigations. Using smaller values for this parameter would enhance the violation of the mass conservation and would make it mandatory to include some diabatic correction to the vertical winds following the procedure described in Rosenlof (1995) and Konopka et al. (2010).
By including other species like CO, ozone, $CH_4$ and water vapor and comparing such distributions with observations, we plan in the future to reduce these degrees of freedom. On the other hand, by covering the whole range of possible variability of tropospheric transport, we will be also able to find the respective variability of the air composition entering the stratosphere.

## Appendix A:  Vertical instability, Brunt-Vaisala frequency and convective available potential energy (CAPE)

Vertical instability is strongly related to the concept of buoyancy. We condense now some textbook knowledge and start from the Boussinesq approximation of the vertical momentum equation by taking into account only buoyancy effects (see e.g. Salby (1996) or Vallis (2006)), i.e.:

$$\frac{Dw}{Dt} = -\frac{1}{\rho_0}\frac{\partial \Delta p}{\partial z} - g\frac{\Delta \rho}{\rho_0} \approx -g\frac{\Delta \rho}{\rho_0} \,. \tag{A1}$$

The solution of this equation describes the vertical velocity $w$ of an air parcel in a hydrostatic reference atmosphere, i.e. defined by the relation $\partial_z p_0 = -\rho_0 g$ with mean pressure and air mass density profiles given by $p_0(z)$ and $\rho_0(z)$, respectively ($z$ - geometric altitude, $g$ - gravity of Earth). The $\Delta$-quantities, i.e., $\Delta p$ and $\Delta \rho$ describe the deviation from such a mean state.

The right hand side of eq. (A1) quantifies the buoyancy. Following Archimedes, the mass-related buoyancy $F_b/m$ can be derived from the weight of the environmental fluid replaced by the parcel, i.e.

$$-\frac{F_b}{m} = \frac{\rho g V - \rho_0 g V}{\rho V} = g\frac{\rho - \rho_0}{\rho} \stackrel{\text{Boussinesq}}{=} g\frac{\rho - \rho_0}{\rho_0} = g\frac{\Delta \rho}{\rho_0}, \tag{A2}$$

where, using the Boussinesq approximation, $\rho$ was replaced by $\rho_0$. With $w = \Delta z/dt$ and $\Delta \rho = -(d\rho_0/dz)\Delta z$, the following equation for $\Delta z$ can be derived:

$$\frac{d^2}{dt^2}\Delta z + N^2 \Delta z = 0, \quad N^2 = -\frac{g}{\rho_0}\frac{d\rho_0}{dz} \,, \tag{A3}$$

with $N$ denoting the Brunt-Vaisala frequency. Using the ideal gas law and assuming the incompressibility of the flow, we get (for details see e.g. Vallis (2006)):

$$\frac{\Delta \rho}{\rho_0} = -\frac{\Delta \theta}{\theta_0} \,. \tag{A4}$$

With $\Delta \rho = -\Delta \theta (\rho_0/\theta_0) = -(\rho_0/\theta_0)(d\theta_0/dz)\Delta z$, the Brunt-Vaisala frequency can be also rewritten to the well-known definition (index "o" is omitted):

$$N^2 = \frac{g}{\theta}\frac{d\theta}{dz} \,. \tag{A5}$$

In general, eq. (A3) has two types of solutions: periodic (i.e. stable) solutions for $N^2 > 0$ and exponentially increasing (i.e. unstable) solutions for $N^2 < 0$. We conclude that the lapse rate of the potential temperature $\theta$ defines the stability and the instability of the atmospheric environment through the positive and negative lapse rate $d\theta/dz$ (or through the positive and negative values of $N^2$), respectively.

Now, we generalize this concept to the atmosphere containing water vapor, i.e. to the moist atmosphere (see e.g. Salby (1996)). First, we define the equivalent potential temperature $\theta_e$ by using the equivalent temperature $T_e$, i.e. the temperature of an air parcel from which all the water vapor has been extracted by an adiabatic vapor-water condensation process (see also Fig. A1):

$$\theta_e = \frac{T_e}{T}\theta \,, \qquad T_e = T + \frac{L_v}{c_p}\mu_w \,. \tag{A6}$$

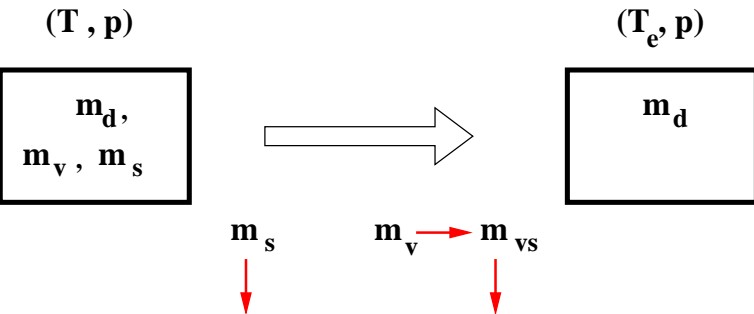

**Figure A1.** The equivalent potential temperature $T_e$ of an air parcel with temperature $T$ and pressure $p$ and with the dry mass $m_d$, water vapor content $m_v$ and the liquid/solid water content $m_s$. $T_e$ is defined by a reversible process completely removing $m_v$ by vapor-water condensation (i.e. transforming pseudoadiabatically $m_v$ into $m_{vs}$) and by using this energy to heat the original air parcel by $p = $ const, i.e. $T_e > T$. Liquid and/or solid water ($m_s$) are removed, but without any correction of $T_e$. (i.e. due to sedimentation this is a pseudoadiabatic and not a pure adiabatic process).

$L_v$ is the latent heat of evaporation and $\mu_w$ the water vapor mixing ratio. The energy released by the phase transition from liquid water to the ice phase can be neglected in most cases because the respective latent heat is smaller by a factor of 10 than the latent heat of gas-liquid transition (334 kJ for melting versus 2270 kJ for evaporation for 1 kg liquid water). There is a number of different definitions of the equivalent potential temperature (Bolton, 1980). Our definition corresponds to the simplified formula proposed by Stull (1988).

Using the same type of arguments as for the dry atmosphere, we also quantify the vertical instability of the moist atmosphere in terms of the lapse rate of the equivalent potential temperature $\theta_e$ or in terms of the respective (moist) Brunt-Vaisala Frequency $N_m$, i.e.

$$N_m^2 = \frac{g}{\theta_e} \frac{d\theta_e}{dz} \; . \tag{A7}$$

However, atmospheric environments with a negative lapse rate of the equivalent potential temperature or with negative values
of $N_m^2$ define only the so-called conditionally unstable atmosphere (see e.g. Salby (1996)), i.e. regions which could be unstable if the respective phase transition releasing latent heat would happen (such air parcels with $N_m^2 < 0$ are not necessarily saturated, so some unresolved motions like adiabatic gravity waves are needed to get saturation).

   For comparison, we also use the known concept of convective available potential energy (CAPE) which can be understood as a different measure of the unstable buoyancy (Emanuel, 1994). Starting form (A2) and (A4), we can write:

$$-\frac{F_b}{m} = g\frac{\rho - \rho_0}{\rho_0} = -g\frac{\theta - \theta_0}{\theta_0} \tag{A8}$$

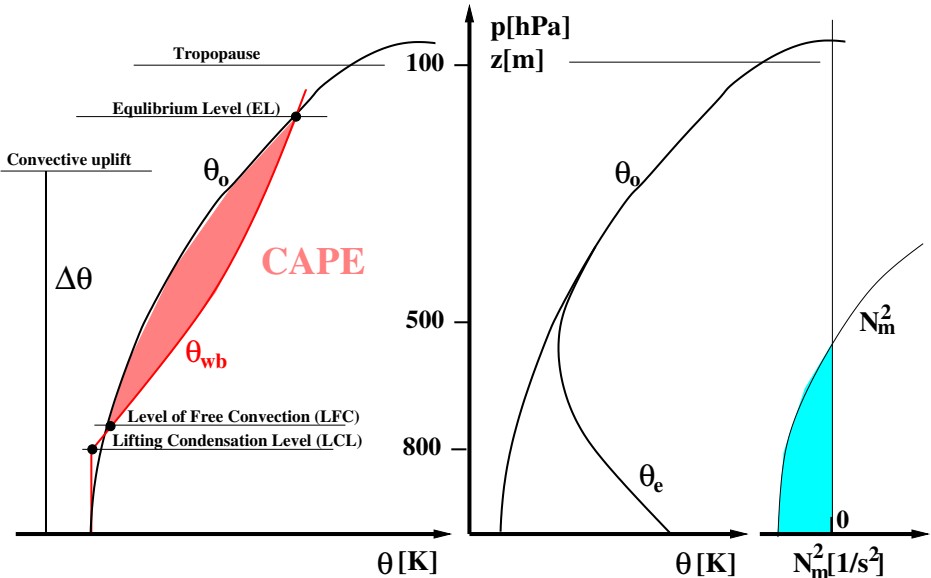

**Figure A2.** CAPE (red) versus stability-based (cyan) measure of potential convective uplift $\Delta\theta$ transporting CLaMS air parcels from the PBL to the upper troposphere. For more explanation see text.

Then, CAPE is defined as the following integral (in J/kg, see also Fig. A2):

$$\text{5} \quad \text{CAPE} = \int\limits_{z_{min}}^{z_{max}} \text{"unstable buoyancy force"} \, dz = \int\limits_{\text{LFC}}^{\text{EL}} g \frac{\theta_{wb}(z) - \theta_0(z)}{\theta_0(z)} \, dz \,. \tag{A9}$$

Whereas $\theta_0(z)$ is the dry potential temperature of the environment (ambient air), $\theta_{wb}(z)$ is the so-called wet-bulb potential temperature which needs some further explanations: We begin with the lifting condensation level (LCL) defined as the height at which a parcel of air becomes saturated when it is lifted adiabatically from the Earth's surface (so the potential temperature does not change). Starting from the LCL the air parcel is than transported along a moist adiabat (also known as saturation-

10 adiabatic process, i.e. an pseudoadiabatic process for which the air is saturated). The corresponding dry potential temperature along such a moist adiabat defines the wet-bulb potential temperature. Note that a distinction is made between the reversible process, in which total water is conserved, and the pseudoadiabatic or irreversible moist adiabatic process, in which liquid water is assumed to be removed as soon as it is condensed (see also Fig. A1). The cross points of such a moist adiabate with $\theta_0(z)$ define the region with unstable conditions (red). The lowest cross point defines the level of free convection (LFC) whereas the equilibrium level (EL) is the height where the (potential) temperature of a buoyantly rising parcel again equals the (potential) temperature of the environment.

In this paper, we use the condition $N_m^2 < 0$ rather than condition CAPE $> 0$ to find those air parcels in the lowest model

level (approximating the PBL) which might undergo convection (see Fig. A2). In the next appendix, we discuss an additional

assumption quantifying the vertical displacement of such air parcels.

## Appendix B:  Latent heat release versus vertical transport

To derive the expression (B7) quantifying the convection-driven vertical uplift $\Delta\theta$ from the latent heat $\delta Q$ available within the

air parcel, we assume that the source of heating is only the latent heat release of water vapor condensation. For a unit mass, we

get along a moist adiabate (see e.g. Salby (1996)):

$$\delta Q = -L_v d\mu_s \tag{B1}$$

where $L_v$ is the specific latent heat for water evaporation (or condensation, in J/kg) and $\mu_s$ is the saturation water vapor mass

mixing ratio.

Using the entropy definition of the potential temperature $\theta := \theta_0 \exp(s/c_p)$ with $s$ being the specific entropy measured in

J/(K kg) (i.e. entropy per unit mass), $c_p$ denoting the specific heat at constant $p$ and $\theta_0$ being the temperature $T_0$ that the air

parcel would acquire if adiabatically brought to the surface pressure $p_0$, we get:

$$ds = c_p \frac{d\theta}{\theta}. \tag{B2}$$

Thus, using the second low of thermodynamics, $ds = \delta Q/T$, and substituting $\delta Q$ by the relation (B1), $d\theta$ can be derived as

$$d\theta = -\frac{L_v \theta d\mu_s}{c_p T}, \tag{B3}$$

where $T$ is the saturation temperature. Furthermore, we assume that during the condensation process, when the saturated air

parcel moves from the initial state $\theta_0$ to the final state $\theta_0 + \Delta\theta$, the saturation temperature $T$ does not change. Thus, eq. (B3)

can be simply integrated, i.e.:

$$\int_{\theta_0}^{\theta_0+\Delta\theta} \frac{d\theta}{\theta} = - \int_{\mu_s(\theta_0)}^{\mu_s(\theta_0+\Delta\theta)} \frac{L_v d\mu_s}{c_p T}. \tag{B4}$$

The result of the integral can be written as:

$$\Delta\theta = \theta_0 [\exp\left(\frac{L_v \Delta\mu_s}{c_p T}\right) - 1], \tag{B5}$$

where $\Delta\mu_s = \mu_s(\theta_0) - \mu_s(\theta_0+\Delta\theta)$. Because $\frac{L_v \Delta\mu_s}{c_p T} \ll 1$, a first order approximation of the right hand side of eq. (B5) through

the Taylor expansion of the exponential function (Ertel, 1938) is

$$\Delta\theta = \frac{L_v \theta_0 \Delta\mu_s}{c_p T}, \tag{B6}$$

which gives a relationship between the total change of the potential temperature and the change of water vapor mass mixing ratio. Strictly, $\Delta\mu_s$ is the change of the water vapor saturation mass mixing ratio before and after a model time step. In our "deep convection" scheme, $\Delta\mu_s$ is estimated by the total water vapor mass mixing ratio $\mu_w$ before the model time step with the assumption that 1) the time scale of deep convection and its associated condensation is smaller than one model time step (here: 6 hours), 2) the residual water vapor content after deep convection ($\mu_s(\theta_0 + \Delta\theta)$) is so small that it can be neglected. Thus, the $\Delta\mu_s$ is assumed to be the total water vapor mass mixing ratio $\mu_w$ in the air parcel within the lowest layer of CLaMS where the criterion ($N_m^2 < 0$) is fulfilled. Therefore, the uplifting of an air parcel in our "deep convection" scheme is estimated by

$$\Delta\theta = \frac{L_v\theta_0\mu_w}{c_pT}. \tag{B7}$$

## Appendix C: Mass flux due to unresolved convective updrafts

To estimate the influence of our convection parametrization on the mass budget, we calculate the zonally-resolved (i.e. per latitude bin $\Delta\phi$), mass density change $\dot{M}_c = dM_c/dt$ within CLaMS layer $\Delta\zeta$ for all convective events which were initiated during one advection time step $\Delta t = 1/4$ day, i.e.:

$$\dot{M}_c(\phi,\zeta) = \sigma(\phi,\zeta)f_m(\phi,\zeta)\frac{\Delta\zeta}{\Delta t} \tag{C1}$$

Units of $\dot{M}_c$ are kg/m$^2$day. The $\zeta$-related air density $\sigma$ is given as $-(1/g)\partial p/\partial\zeta$ (units: kg/Km$^2$). Furthermore, $f_m$ is the fraction of air parcels convectively transported within the latitude bin $\Delta\phi$ to a target layer $\Delta\zeta$ denoted here as $n_{conv}$ relative to the total number number of air parcels in this destination region denoted here as $N_{tot}$, i.e. $f_m = n_{conv}/N_{tot}$ (we assume that all air parcels are uniformly distributed within the layer, both horizontally and vertically). The related convective updraft mass flux density, $F_c$, is given through $\partial_z F_c = \dot{M}_c$ or as a discretized solution:

$$F_c(\phi_i,\zeta_j) = \sum_{k=j}^{k=N} \dot{M}_c(\phi_i,\zeta_k) \tag{C2}$$

where $N$ denotes the total number of CLaMS layers an $\phi_i$, $i = 1,\ldots,M$ are the latitude bins.

Fig. C1 shows the annually averaged values of $F_c$ (a) and its global mean (b). The latter profile quantifies the additional mass flux into the model caused by parametrized convective updrafts and it roughly balances the mass deficit diagnosed in Fueglistaler et al. (2009) (c.f. their Figure 10) due to use of the non mass-conserving diabatic vertical velocities.

*Author contributions.* P. Konopka and M. Tao conceived most of the presented ideas and performed the numerical simulations. P. Konopka wrote the manuscript with support from F. Ploeger. M. Diallo helped to use the CONTRAIL and the CarbonTracker data. M. Riese supervised the findings of this work.

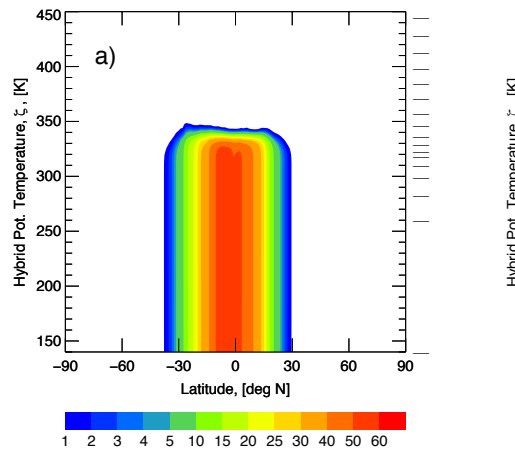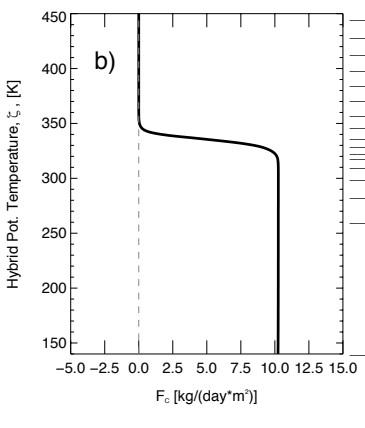

**Figure C1.** Annually averaged and zonally resolved updrafts due to parametrized convection ($\sigma_r = 0.3$) (a) and their global mean (b). On the right side of panel (a) the boundaries of CLaMS layers, $\Delta\zeta_i$, $i = 1 \ldots N$, are shown.

*Acknowledgements.* The European Centre for Medium-Range Weather Forecasts (ECMWF) provided meteorological analysis for this study. We thank to the CONTRAIL team, in particular to Toshinobu Machida and Yousuke Sawa for helping us to use this data set. The authors sincerely thank Andy Jacobson and Pieter Tans from NOAA for support related to the CarbonTracker data. We also thank Rolf Müller and Michael Volk for helpful discussions. We are thankful to Marta Abalos for providing us with the WACCM e90 climatology. Excellent programming support was provided by N. Thomas. J.-U. Grooß helped us to make the source code available at the GitLab server. Finally, we would like to thank all 2 reviewers for their insightful and probably very time-consuming reviews, as these comments led us to an improvement of the work. Especially, we thank Ingo Wohltmann for his important comment related to the mass budget violation if unresolved convective updrafts are considered. This research was supported by the German Helmholtz-Gemeinschaft within the Helmholtz-CAS joint research group (JRG) "Climatological impact of increasing anthropogenic emissions over Asia".

*Code and data availability.* CLaMS v2.0 discussed in this paper is available at the GitLab server: https://jugit.fz-juelich.de/clams/CLaMS as well as at the online repository Zenodo with DOI: 10.5281/zenodo.2632683 (search for "CLaMS 2.0" with the Zenodo search engine). Version 2.0 can be reduced to the version 1.0 (Pommrich et al., 2014) by switching off the vertical mixing and convective updrafts (in the main CLaMS script). The CLaMS v1.0 code is now implemented into the Modular Earth Submodel System (MESSy) system (Version 2.54 at the Mercurial server messy.fz-juelich.de/messy-2.54.0-clams).The here described version 2.0 will be included into the one of next releases of MESSy.

The CONTRAIL $CO_2$ data are openly accessible ( doi:10.17595/20180208.001). The CarbonTracker data (version 2013B) can be downloaded from ftp:/products/carbontracker/co2/CT2013B/molefractions/co2_total/. For more detailed model data, please contact the authors.

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
