# Peer review of "Tropospheric mixing and parametrization of unresolved convective updrafts as implemented into the Chemical Lagrangian Model of the Stratosphere (CLaMS v2.0)"

_Geoscientific Model Development, 2018_

## Short Comment (SC1) · 31 Jul 2018

Dear authors,

In my role as Executive editor of GMD, I would like to bring to your attention our Editorial version 1.1:

http://www.geosci-model-dev.net/8/3487/2015/gmd-8-3487-2015.html

This highlights some requirements of papers published in GMD, which is also available

on the GMD website in the 'Manuscript Types' section:

http://www.geoscientific-model-development.net/submission/manuscript_types.html

In particular, please note that for your paper, the following requirements have not been met in the Discussions paper:

- "The main paper must give the model name and version number (or other unique identifier) in the title."

- "All papers must include a section, at the end of the paper, entitled 'Code availability'. Here, either instructions for obtaining the code, or the reasons why the code is not available should be clearly stated. It is preferred for the code to be uploaded as a supplement or to be made available at a data repository with an associated DOI (digital object identifier) for the exact model version described in the paper. Alternatively, for established models, there may be an existing means of accessing the code through a particular system. In this case, there must exist a means of permanently accessing the precise model version described in the paper. In some cases, authors may prefer to put models on their own website, or to act as a point of contact for obtaining the code. Given the impermanence of websites and email addresses, this is not encouraged, and authors should consider improving the availability with a more permanent arrangement. After the paper is accepted the model archive should be updated to include a link to the GMD paper."

Therefore please add the version number of CLAMS in the title of your publication upon revision.
Additionally, the code availability section is really confusing. So what is your publicatoin about? Did you used CLAMS or CLAMS/MESSy? Please note, that the Code Availability Section primarily refers to the exact version, which is published in the article. So if you used CLAMS (without MESSy) so please state how to access CLAMS. If you

used CLAMS/MESSy state it in the article itself. Of course, you are free to add references the further developed version (here CLAMS/MESSy) in the Code Availability section. In that case, please state in which version of MESSy the new updates will be available and please add more information on how to access MESSy and especially CLAMS/MESSy.

Yours,

Astrid Kerkweg

---

## Referee Comment (RC1) · Anonymous Referee #1 · 20 Aug 2018

This study introduces a heuristic idea to extend the mixing scheme in CLaMS down into the troposphere. Konopka et al. parametrised an additional mixing process using the dry Richardson number and added it to the existing CLaMS mixing scheme. Moreover, convective uplift in the troposphere is represented by using the moist Richardson number to calculate an additional vertical displacement to the trajectory. Konopka et al. evaluate their new parametrisations by comparison of Age-of-Air for the stratosphere, propagation of $CO_2$ from the planetary boundary layer to the lower stratosphere, and $CO_2$ variability with observations.

The paper is well written and clear structured and is suitable for publication in GMD after revision.

General Comments:

1. Title: The term "parametrisation of convection" suggests, that you describe a more or less comprehensive convection parametrisation with up- and downdrafts and compensation motion. But what you actually show (you mentioned it in the abstract) is "convective uplift and mixing". Please, modify the title accordingly.

2. The citation of other publications describing Lagrangian models, which already use Lagrangian convection is missing. Please, cite other work introducing Lagrangian convection or convective mixing, e.g. Collins et. al. (2002) Q.J.R.M., Erukhimova and Bowman (2006), J. Geophys. Res., Forster et al. (2007), J. Appl. Meteorol. Clim.

3. I would suggest to select the word " simulations" instead of "runs".

4. Vaisala or Väisälä? Decide for one notation.

Specific comments:

Page 2, line 18: $Ri_c$: omit the subscript here.

Page 2, line 28: "considered" instead of "detected"

Page 2, line 30: Instead of "roughly" just say "above"

Page 2, line 32: heuristic (not heurestic)

Page 3, line 34: reasonable

Page 3, line 35: Please specify here, at what height your lowest model boundary is. From your Figures I would suggest that it is the surface layer?

Page 3, line 38: upward

Page 3, line 7: Please explain: is there a reason why you compare a one year mean of
2007 (CLaMS) with a WACCM climatology over 1955-2099? Is the WACCM simulation of AMIP type or a time-slice experiment?

Page 4, line 5-7: What did you wanted to say with this sentence?

Page 5, line 3: The point at the end of the sentence is missing.

Page 5, line 6: "condition"

Page 6, line 17: In Table 1 you list 2 reference simulations. Please use the same abbreviation in the text. Is REF or REF-6h meant by the reference simulation?

Page 6, line 22: You refer to a period between 2005-2008 of the reference simulation. Please explain, how do you calculate Age-of-Air from such a short simulation? I would expect that at least a 20-year model simulation had to be performed to calculate Age-of-Air.

Page 7, line 11: Vertical boundaries of layers on the left side? What do the tics on the right side of Figure 7 mean?

Page 11, line 12: Instead of "validate" please say "evaluate".

Page 12, line 2: You refer to a relatively short period of the control simulation (2005-2008). Please explain, how you calculate Age-of-Air from such a short simulation.(Figure 7 and 11)

Page 13, Fig.7: Please use a colour bar, which better shows the differences of Age-of-Air in the troposphere, because your modifications in CLaMS have only less influence on the stratospheric distribution of Age-of-Air. Moreover, the influence of the reference pressure on the results in the troposphere should be better visible for a discussion. Alternatively, you might also present the difference of the (control-300hPa) minus (control-700hPa) simulation. In Table 1 you list the simulations TROP_MIX and UNRES_CONV, representing simulations with either tropospheric mixing or convective uplift. Could you please show the result of Age-of-Air for these simulations, also? This

would provide more information on the spacial impact of different mixing attempts on the troposphere.

Page 14, Figure 8: Please describe the simulations you analysed in the Figure caption. 8a-d. 8cd) seems to be the reference simulation, but it is not clear, if it is REF or REF-6h.

Page 15, line 22-25: Please explain, why it is plausible, that you find a faster upward propagation in the CLaMS control simulations compared to REF. Why is the interhemispheric transport weaker in the sigma=0.7 simulation?

Page 16, Figure 9: The date in the Figure caption should be 25.9.

Page 19, Figure 11: 11c) shows a simulation not described in Table 1. I found 3 simulations with the same name (FULL_EXT), but with different set-ups. Are 11b+c) simulations with the reference pressure at 300hPa or 700hPa?

Page 18, 10-10: Please use the abbreviations for the simulations TROP_MIX and UN-RES_CONV also in the text. So far they can only be found in Figure 10 and Table 1.

Page 18, line 5: "Although there" instead of "Also there"?

Page 18, line 1-5: If you mention simulations, please refer to the notations in Table 1. All control simulations should be described in section 4.1 and summarised in Table 1.

Page 20, line 3-5: The last part of the sentence "especially the effects on the stratospheric water vapour" makes no sense.

Page 20, line 26: temperature

Page 21, Figure caption A1: removed,

Page 24, line 7: air parcel

---

## Short Comment (SC2) · 15 Nov 2018

Dear authors,

first I have to apologize for the late comment, I just stumbled over this interesting manuscript at the last minute.

Given that your method is so different from the usual method of parameterizing convective transport in Lagrangian models (e.g. Collins et al., QJRMS, 128, 991-1009, 2002, Forster et al., J. Appl. Meteorol. Clim., 46, 403-422, 2007, Rossi et al., Geosci.

Model Dev., 9, 789-797, 2016), I wonder about 2 things.

1. Do you get reasonable convective mass fluxes or detrainment and entrainment rates from your approach? The schemes cited above all take a statistical approach which uses given convective mass fluxes and entrainment and detrainment rates. E.g., typically a probability for entrainment of a Lagrangian air parcel into convection is calculated by comparing the mass of air entrained in a given time step (according to the entrainment rate in a layer) to the mass of that layer. Then, random numbers are drawn to determine if a given trajectory goes into convection. This approach implicitly makes sure that the convective mass fluxes, entrainment rates and detrainment rates from the underlying convection parameterization (e.g. Tiedtke parameterization in ERA Interim) are correctly reproduced (this requires averaging over a large enough ensemble of trajectories). It also considers that the mass associated with an air parcel in a global Lagrangian model is usually much larger than the mass transported in a single convective event. As far as I can see, you don't mention convective mass fluxes or entrainment or detrainment rates in your manuscript, and there is no constraint in your method on the convective mass fluxes. I think it would be interesting to discuss this (even considering that you can't normally compare to measurements (exceptions are few, as e.g. Kumar et al., J. Atmos. Sci., 72, 1837-1855, 2015), but maybe comparing to the convective mass fluxes and detrainment rates of ERA Interim would be nice).

2. The updraft of the air parcels in convective events needs to be balanced by subsidence in cloud free air to ensure mass conservation. Usually, Lagrangian convective transport schemes do contain an explicit parameterization for this. It seems that you don't consider subsidence explicitly, and that you rely on the CLaMS mixing scheme to handle that "automatically" ("the mixing procedure is able to adjust a certain increase or decrease in the number of air parcels", page 11, line 1). This seems to be an unrealistic approach at first glance. Wouldn't it be better to subside all air parcels outside convection by a given distance, as it is done in other schemes?

Best regards, Ingo Wohltmann

---

## Short Comment (SC3) · 26 Nov 2018

Our intention for parameterization of convection is the simplest possible approach to study the global impact of this effect on tracer distributions, especially on the upward propagation of the $CO_2$ annual cycle. We thank Ingo Wohltmann for his comment and agree, that in the current version, this parameterization is not mass-conserving, however there are few arguments (listed below) which support our simple method:

1. The total mass transported through every potential temperature surface should

[Figure]

vanish at least in the annual mean. The residuum of the total mass transported through hybrid potential temperature surfaces extending between the Earth's surface and 400 K pot. temperature level was calculated for every 6-hours time step of the ERA-Interim data (exemplary for 2005) and is shown below (in contrast to pot. temperature, the hybrid potential temperature surfaces do not cross the Earth).

Thus, the total budget shows a deficit (blue values) roughly in the region between 700 and 200 hPa. Our explanation for this deficit is that the horizontal resolution of this data (roughly 80 km) does not sufficiently resolves the convective towers (which are of the order 1 km) but does better resolve the large-scale convective downdrafts which, mainly due to radiatively active water vapor, are better reproduced in the ERA-Interim vertical velocities (i.e. mainly in their diabatic part). Thus, our parameterization aims to close this in-balance by including some additional convective updrafts. The results show that qualitatively our simple approach closes (or slightly reverses) such a deficit. That means, that we mainly include the convective updrafts to minimize such a deficit rather than to formulate a mass-conserving parameterization of convection. We will discuss this point in the revised version more clearly.

2. Our convective updrafts from the lowest model layer (approximating the PBL) are "in principle" balanced by the reverse downdraft transporting a CLaMS air parcel from the position where the convective updraft "ends" back to the PBL. However, because we overwrite all the mixing ratios in the PBL by a prescribed boundary condition, this downdraft does not have any influence on the tracer distributions in the here presented CLaMS version (so this is the reason for the wording "in principle"). Here, we follow much simpler Lagrangian ideas like in Ueyama et al., JGR, 2018 to include the effect of convection only on the upward transport of tracers.

3. Finally, CLaMS is using diabatic vertical velocities following procedure described

in Mahowald et al., JGR, 2002, which are *per se* not mass-conserving. Whereas above $p_r = 300$ hPa pure diabatic velocities are used, below $p_r$ a hybrid vertical velocity is applied combining the kinematic (mass-conserving) with the diabatic (non mass-conserving) velocity. Also the CLaMS mixing scheme itself is not mass-conserving in a strict mathematical sense. However, it should be not an excuse to use a better mass-conserving convection scheme in the future.

We conclude that we certainly have to improve the explanation of our procedure in the revised version of our paper. Following the spirit of the published schemes proposed in the comment, we plan in the future to include such a better, mass-conserving scheme. Nevertheless, the presented approach significantly improves the performance of CLaMS in the troposphere and gives some first insights on the importance on convection, so we still believe that such a simplified approach has its justification.

―――――――――――――――――

[Figure]

[Figure]

**Fig. 1.** Residuum of the total mass flux calculated at every hybrid potential temperature level surface extending between the Earth's surface on the 400 K pot. temperature level (ERA-Interim, 2005)

---

## Referee Comment (RC2) · Anonymous Referee #2 · 15 Dec 2018

Reviewer (Comments):
**Review of " Tropospheric mixing and parametrization of unresolved convection as implemented into the Chemical Lagrangian Model of the Stratosphere (CLaMS)" by Paul Konopka et al.**

**Recommendation: Publication after revision**

The paper is very well organised and written. The topic discussed in this paper, "Improving the state-of-the-art Lagrangian transport model for the stratosphere CLaMS by extending the transport scheme to the troposphere", is in general of high relevance. The reason is, that this would be one important step to enable climate and air quality modelling with the fully Lagrangian CLaMS transport scheme. The simulations with the extended CLaMS transport scheme are validated against satellite observations of the Outgoing Longwave Radiation (OLR) and in-situ observations of $CO_2$ in the UT/LS by CONTRAIL (Comprehensive Observation Network for TRace gases by AIrLiner observations).

The paper should be submitted after addressing the comments below.

**General comments:**

The goal to improve the tropospheric tracer transport has been reached with the extended CLaMS transport scheme using a heuristic approach for a better representation of tropospheric mixing and unresolved convection. This result is, as outlined above, very valuable and implicates a high potential for future application of the CLaMS transport scheme. However, to my point of view there are some things missing or at least should be better explained or motivated.

1.) *Mass conservation in CLaMS*
For this topic, the focus is on the new parameterisation of unresolved convection, because the new tropospheric mixing scheme does not change the redistribution of mass compared to the actual reference version of CLaMS. Although, it is relevant to understand what the adaptive regridding is doing in terms of mass conservation. In this context, I highly appreciate the comments by Ingo Wohltmann and the conclusion by the authors to his comments: "We conclude that we certainly have to improve the explanation of our procedure in the revised version of our paper."
My suggestion would be to use the mass flux residuum of ERA-Interim as shown in Fig.1 in the answer to Ingo Wohltmann's comments motivating their heuristic approach (in section 1 or 2). Additionally, it would be very interesting to see also, how the mass flux residuum looks for the CLaMS reference simulations with standard adaptive regridding and for the control simulations with additional convective uplift (in section 4). The latter should show that the deficit in the range of 700 to 200 hPa has been reduced significantly.

2.) *Validation of UTLS transport in CLaMS with in-situ $CO_2$ observations (CONTRAIL)*
$CO_2$ and especially the propagation of its seasonal cycle from the PBL into the UTLS is highly useful for model transport validation. Here, the authors use as benchmarks the in-situ $CO_2$ measurements by CONTRAIL aboard of passenger aircraft and the assimilated $CO_2$ data set provided by CarbonTracker. The latter is mainly constraint by surface measurements (see Table 1 in https://www.esrl.noaa.gov/gmd/ccgg/carbontracker/). Therefore, the CONTRAIL data are the reference for $CO_2$ in the UTLS and the CarbonTracker data are the reference for the PBL.

My criticism is the way the CONTRAIL data are used for the evaluation in section 4.3. The CONTRAIL $CO_2$ measurements are zonally and monthly averaged between 2005 and 2008 and interpolated at a latitude-altitude grid with 10° by 1 km resolution and extending between 20°S to 60°N and 5.5 to 12.5 km. In the extratropics, this approach is highly problematic, because the mean (or the interpolated) $CO_2$ mixing ratios in the grid boxes will be strongly affected by (the irregular and sparse) sampling, especially above 7-8 km, because the individual probed air mass could be tropospheric or stratospheric. This matters for the months when cross-tropopause gradients are large, i.e. February to May in the NH. Also, the seasonal cycle of $CO_2$ is quite different below and above the extratropical tropopause. My suggestion is to filter out the stratospheric CONTRAIL $CO_2$ data to avoid this issue. This would not weaken the evaluation, because the $CO_2$ seasonal cycle in the free and upper troposphere is the relevant diagnostic for the introduced new tropospheric transport scheme in CLaMS. Also the representation of the $CO_2$ seasonal cycle in the tropical, subtropical (shown exemplarily) and extratropical free and upper troposphere should be discussed in a bit more detail. It is a known issue that modelled tracer transport from the PBL into the extratropical UT is often too weak, especially during summer.

**Specific comments:**

p.2, l.30: I think you mean here "…above the level of 300 hPa…" and not "…about 300 hPa…"

p.2, l.31-33: Why you use only the nominator – the buoyant production of turbulence – of the gradient Richardson number Ri to parametrise tropospheric instabilities and not Ri itself?

p.3, l.34-38: Just for curiosity, would the actual CLaMS transport scheme accumulate tracer in the PBL, if one uses emissions instead of prescribed PBL mixing rations?

p.4, l.5-7: "*Although we are aware of numerous convective schemes (e.g. Tiedtke (1989); Emanuel (1991)), our approach mainly intends to cover the range of possible variability due to unresolved tropospheric transport.*"
It is quite unclear to me what is meant here, see also point 1.) in the general comments.

p.5, Figure 2: It seems that the TIL in the extratropics is (very) weakly pronounced compared to e.g. Birner et al. (2006). Is there an explanation?

Birner, T., D. Sankey, and T. G. Shepherd (2006), The tropopause inversion layer in models and analyses, Geophys. Res. Lett., 33(14), doi:Doi 10.1029/2006gl026549.

p.7, l.23-26: How the free parameter Nc, the critical limit for static stability, below which enhanced tropospheric mixing is triggered in the new parameterisation, is estimated? Which Nc value has been used for the CLaMS control simulations? How sensitive are the results to the choice of Nc?

p.8, l.1-2: I understand, that it is technically easier and better comparable to the previous CLaMS version, if the step of the additional tropospheric mixing is executed after the adaptive regridding, but does this also makes sense from a physical point of view? Would the result be different with the inverse transport operator: First step: mixing due to vertical instability and second step: mixing due to strong horizontal wind shear (deformation)?

p.9, Figure 4: A similar question: Why you implemented the vertical displacement in your new parameterisation for unresolved deep convection after the horizontal displacement? Is this realistic for deep convection?

p.9, l.11: Same question as for Nc: How sensitive are the results to the choice of the criteria $N^2_m < 0$ triggering convective events in the parameterisation?

p.11, l.1-2: "*It means that the mixing procedure is able to adjust a certain increase or decrease in the number of air parcels, but this amount should be below ± 10%.*"
This has to be better explained, see point 1.) in the general comments.

p.11, l.1-2: Again, how sensitive are the results of the control simulations to the choice of criteria that only $\Delta$Theta > 35 K triggers convection? Would it not be better to use instead of a fix value the criteria Theta+$\Delta$Theta > upper level of the PBL? This would mean that only convective events are considered that lift the air parcel out of the PBL, where the $CO_2$ mixing ratios of the air parcel will be overwritten by the prescribed value in the next time step anyway.

p.12, l.17-20: "*It should be emphasized…*"
What would this mean for simulations using emission fluxes rather than prescribed surface mixing ratios (see also my question p.3, l.34-38 above)?

p.13, Table 1: The critical values of the dry and moist Brunt Vaisala frequency used for the simulations should be added here. Also, it would be easier to use different names for both simulations FULL_EXT (e.g. FULL_EXT_0.3 and _0.7).

p.14, Figure 8: Please specify the simulations:
Top: FULL_EXT_0.3 or FULL_EXT_0.7?
Bottom: REF or REF-6h?

p.15, l.4: Boucher et al. (2009) is not really a good citation for stratospheric $CO_2$ and methane oxidation. The chemistry of methane in the middle atmosphere was to my knowledge first considered by Bates and Nicolet (1950). Early measurements of stratospheric $CO_2$ and $CH_4$ profiles date back to the 1960s and 70s, e.g. by Ehhalt.

Bates, R. D., and M. Nicolet (1950), Atmospheric Hydrogen, Publications of the Astronomical Society of the Pacific, 62(365), 106.

Ehhalt, D.H., The atmospheric cycle of methane. Tellus: 26, 58, 1974.

Ehhalt, D.H., L.E. Heidt, R.H. Lueb, and E.A. Martell, Concentrations of CH4, CO, CO2, H2, H2O and N2O in the upper stratosphere. J Atmos Sci: 32, 163, 1975.

p.15, l.16: How the reference simulations has been initialised on 1.1.2000? With an empty atmospheric domain – no $CO_2$? If so, the spin-up of only 5 years might be a bit too short for UTLS $CO_2$ analysis.

p.15, l.16: "*… all other control runs were started using the output of the reference run for the initial distribution.*"
This is a bit unclear to me. Does this mean that the control runs have started with the output of the reference run from 31.12.2004? If so, the results of the control runs for 2005, at least for the LS, is influenced by the prescribed distribution of the reference simulation. In the UT, this should only influence the first months of 2015, but still, this is not optimal for the comparison.

p.15, l.25: "*…on, the former being in better agreement with CarbonTracker.*"

I think, this statement is a bit misleading, because $CO_2$ assimilated from CarbonTracker data set is not the reference for the middle troposphere or UTLS, see also general comment point 2.).

p.17, Figure 10: Should be improved, see general comment point 2.).

p.17, Figure 10, Legend: According to Table 1, the name of the data set in the legend should be TROP_MIX and not VERT_MIX.

p.17, Figure 10, Caption: The altitude range in (b) is 10.5 km and not 15.5 km.

p.17, l.3-5: For the extratropics the gridding of the aircraft data has to be done in tropopause related coordinates or has to be filtered for tropospheric data, see general comment point 2.). Otherwise the averages are strongly biased by the sampling statistic of tropospheric and stratospheric air in the individual bins.

p.17, l.5-6: Has CLaMS simulations been sampled along the flight track?

Sec.4.4, p.18, l.10-11: This should be simply demonstrated by comparing REF vs. REF-6h.

Sec.4.4, p.18, l.16-18: This might be true, but I cannot really see the differences in AoA between Fig 11b and c. To demonstrate this, a difference plot would be necessary.

---

## Author Comment (AC1) · 25 Feb 2019

Response to reviewer 1

We would like to thank reviewer 1 for a very thoughtful and detailed review of our manuscript that helped to improve the paper. In the following, we address all the points raised in the review (denoted by italic letters). Our responses are highlighted in blue. The references to the attached manuscript as well as substantial changes in the manuscript are highlighted in red.

**General comments:**

1. *Title: The term "parametrisation of convection" suggests, that you describe a more or less comprehensive convection parametrisation with up- and downdrafts and compensation motion. But what you actually show (you mentioned it in the abstract) is "convective uplift and mixing". Please, modify the title accordingly.*
   Yes, we completely agree, so we changed our title following the recommendation.

2. *The citation of other publications describing Lagrangian models, which already use Lagrangian convection is missing. Please, cite other work introducing Lagrangian convection or convective mixing, e.g. Collins et. al. (2002) Q.J.R.M., Erukhimova and Bowman (2006), J. Geophys. Res., Forster et al. (2007), J. Appl. Meteorol. Clim.*
   All these references are now included. Thanks a lot for these important citations.
   P4, L10-14

3. *I would suggest to select the word "simulations" instead of "runs".*
   ...was done

4. *Vaisala or Väisälä? Decide for one notation.*
   We changed everywhere to Vaisala.

**Specific comments:**

1. *Page 2, line 18: $Ri_c$: omit the subscript here.*
   ...was done

2. *Page 2, line 28: "considered" instead of "detected"*
   ...was done

3. *Page 2, line 30: Instead of "roughly" just say "above"*
   ...was done

4. *Page 2, line 32: heuristic (not heurestic)*
   ...was done

5. *Page 3, line 34: reasonable*
   ...was done

6. *Page 3, line 35: Please specify here, at what height your lowest model boundary is.*
   ...was done: (thickness: 1.1-1.3 km)

7. *From your Figures I would suggest that it is the surface layer?*
We explain now all the details of the definition of the lowest model layer in the rewritten section 4.1
P13, L16-19, P14, L4-8

8. *Page 3, line 38: upward*
...was done

9. *Page 3, line 7: Please explain: is there a reason why you compare a one year mean of 2007 (CLaMS) with a WACCM climatology over 1955-2099? Is the WACCM simulation of AMIP type or a time-slice experiment?*
The WACCM climatology over 1955-2099 is published in Abalos 2017, see their Fig. 1. The year-to-year variability in the WACCM simulation is much smaller if compared with the deviation of CLaMS, so it is enough to compare CLaMS with the climatology. The respective text was included
P4, L2-4

10. *Page 4, line 5-7: What did you wanted to say with this sentence?*
This paragraph is reformulated now
P4, L10-16

11. Page 5, line 3: The point at the end of the sentence is missing.
...was included

12. Page 5, *line 6: "condition"*
...was included

13. *Page 6, line 17: In Table 1 you list 2 reference simulations. Please use the same abbreviation in the text. Is REF or REF-6h meant by the reference simulation?*
We unified all abbreviations which are listed in Table 1 and use only these abbreviations both in text and in the figures

14. *Page 6, line 22: You refer to a period between 2005-2008 of the reference simulation. Please explain, how do you calculate Age-of-Air from such a short simulation? I would expect that at least a 20-year model simulation had to be performed to calculate Age- of-Air.*
Both reviewers asked for more details describing the details of our runs. For this we put these informations into our revised text: see first paragraphs in section 3.1 and 4.1
P6 L26-30, P13, L9-15

15. *Page 7, line 11: Vertical boundaries of layers on the left side? What do the tics on the right side of Figure 7 mean?*
...it should be "on the right side". We reformulated our text.

16. *Page 11, line 12: Instead of "validate" please say "evaluate".*
...was done

17. *Page 12, line 2: You refer to a relatively short period of the control simulation (2005- 2008). Please explain, how you calculate Age-of-Air from such a short simulation.(Figure 7 and 11)*
See above our answer to the specific comment 14

18. *Page 13, Fig.7: Please use a color bar, which better shows the differences of Age-of-Air in the troposphere, because your modifications in CLaMS have only less influence on the stratospheric distribution of Age-of-Air. Moreover, the influence of the reference pressure on the results in the troposphere should be better visible for a discussion. Alternatively, you might also present the difference of the (control-300hPa) minus (control-700hPa) simulation. In Table 1 you list the simulations TROP_MIX and UNRES_CONV, representing simulations with either tropospheric mixing or convective uplift. Could you please show the result of Age-of-Air for these simulations, also? This would provide more information on the spacial impact of different mixing attempts on the troposphere.*
We changed the color bar in Fig. 7 and we completed Fig. 11 following the recommendation. In this way, the reader can see step by step (Fig. 11b to f), how our changes of transport can be seen in the AoA distribution.
P20 L8 - P21 L17

19. *Page 14, Figure 8: Please describe the simulations you analyzed in the Figure caption. 8a-d. 8cd) seems to be the reference simulation, but it is not clear, if it is REF or REF- 6h.*
The notation was completed following the definition in Table 1

20. *Page 15, line 22-25: Please explain, why it is plausible, that you find a faster upward propagation in the CLaMS control simulations compared to REF. Why is the inter- hemispheric transport weaker in the sigma=0.7 simulation?*
We can only speculate that in the sigma=0.7 simulations the diabatic transport is less noisy than the more kinematic transport in the sigma=0.3 case. We slightly reformulated our text but avoid any type of speculations (following the recommendation of the reviewer 2)

21. *Page 16, Figure 9: The date in the Figure caption should be 25.9.*
...was changed

22. *Page 19, Figure 11: 11c) shows a simulation not described in Table 1. I found 3 simulations with the same name (FULL_EXT), but with different set-ups. Are 11b+c) simulations with the reference pressure at 300hPa or 700hPa?*
The notation was corrected

23. *Page 18, 10-10: Please use the abbreviations for the simulations TROP_MIX and UN-RES_CONV also in the text. So far they can only be found in Figure 10 and Table 1.*
...was done

24. *Page 18, line 5: "Although there" instead of "Also there"?*
....was changed

25. *Page 18, line 1-5: If you mention simulations, please refer to the notations in Table 1. All control simulations should be described in section 4.1 and summarized in Table 1.*
was done

26. *Page 20, line 3-5: The last part of the sentence "especially the effects on the stratospheric water vapor" makes no sense.*
The second part of the sentence was removed.

27. *Page 20, line 26: temperature*
    ...was done

28. *Page 21, Figure caption A1: removed*
    ..was done

29. *Page 24, line 7: air parcel*
    was done

[revised manuscript text omitted]

2, 153–173, 2009.

---

## Author Comment (AC2) · 25 Feb 2019

Response to reviewer 2

We would like to thank reviewer 2 for a very thoughtful and detailed review of our manuscript that helped to improve the paper. In the following, we address all the points raised in the review (denoted by italic letters). Our responses are highlighted in blue. The references to the attached manuscript as well as substantial changes in the manuscript are highlighted in red.

**General comments:**

1. *Mass conservation in CLaMS: For this topic, the focus is on the new parameterisation of unresolved convection, because the new tropospheric mixing scheme does not change the redistribution of mass compared to the actual reference version of CLaMS. Although, it is relevant to understand what the adaptive regridding is doing in terms of mass conservation. In this context, I highly appreciate the comments by Ingo Wohltmann and the conclusion by the authors to his comments: "We conclude that we certainly have to improve the explanation of our procedure in the revised version of our paper". My suggestion would be to use the mass flux residuum of ERA-Interim as shown in Fig.1 in the answer to Ingo Wohltmann's comments motivating their heuristic approach (in section 1 or 2). Additionally, it would be very interesting to see also, how the mass flux residuum looks for the CLaMS reference simulations with standard adaptive regridding and for the control simulations with additional convective uplift (in section 4). The latter should show that the deficit in the range of 700 to 200 hPa has been reduced significantly.*
   We completely agree with this criticism. See for this the revised section 3.3 (first two and last three paragraphs, Figure 4) and appendix C. We also thank Ingo Wohltmann in the acknowledgments for his contribution.
   P9 L2-13, P10 L13 - P12 L33, P26 L12-25

2. *Validation of UTLS transport in CLaMS with in-situ CO2 observations (CONTRAIL) CO2 and especially the propagation of its seasonal cycle from the PBL into the UTLS is highly useful for model transport validation. Here, the authors use as benchmarks the in-situ CO2 measurements by CONTRAIL aboard of passenger aircraft and the assimilated CO2 data set provided by CarbonTracker. The latter is mainly constraint by surface measurements (see Table 1 in ...). Therefore, the CONTRAIL data are the reference for CO2 in the UTLS and the CarbonTracker data are the reference for the PBL.*

   *My criticism is the way the CONTRAIL data are used for the evaluation in section 4.3. The CONTRAIL CO2 measurements are zonally and monthly averaged between 2005 and 2008 and interpolated at a latitude-altitude grid with 10 by 1 km resolution and extending between 20S to 60N and 5.5 to 12.5 km. In the extratropics, this approach is highly problematic, because the mean (or the interpolated) CO2 mixing ratios in the grid boxes will be strongly affected by (the irregular and sparse) sampling, especially above 7-8 km, because the individual probed air mass could be tropospheric or stratospheric. This matters for the months when cross-tropopause gradients are large, i.e. February to May in the NH. Also, the seasonal cycle of CO2 is quite different below and above the extratropical tropopause. My suggestion is to filter out the stratospheric CONTRAIL CO2 data to avoid this issue. This would not weaken the evaluation, because the CO2 seasonal cycle in the free and upper*

*troposphere is the relevant diagnostic for the introduced new tropospheric transport scheme in CLaMS. Also the representation of the CO2 seasonal cycle in the tropical, subtropical (shown exemplarily) and extratropical free and upper troposphere should be discussed in a bit more detail. It is a known issue that modeled tracer transport from the PBL into the extratropical UT is often too weak, especially during summer.*

Here, we also agree but we are not able to do all these calculations in the revised version because this is beyond the scope of this paper. We will certainly follow these ideas in our next paper. In the meantime few comments:

(a) In Figure 10a and b, we added the standard deviation for all CONTRAIL gridded points (dot black points with vertical black lines). In this way, the variability of the gridded data can be seen. A little surprising, the variability at the lower level (5.5 km) is higher than at the higher level (10.5) where a disadvantage of our simple gridding can be expected where tropospheric *and* stratopsheric values may contribute.

(b) We also completely agree that tropopause-based coordinates would be a much better approach but, in the moment, we cannot complete this task in due time. On the other hand, the intention of this paper is to present the extension of tropospheric transport in CLaMS and of the way how it can be validated at "a global scale" by using the upward propagation of the CO2 seasonal cycle. Thus, we would like to shift a more detailed validation of all relevant critical parameters to our next paper.

(c) We also agree, that a comparison of the seasonal cycle of CO2 with the model should be done for different regions of the atmosphere (tropics, subtropics, extratropics, polar regions) This is also something that we will complete in the future. Note also a poor coverage of CONTRAIL observations in the tropics.

**Specific comments:**

1. *p.2, l.30: I think you mean here "...above the level of 300 hPa...´´ and not "...about 300 hPa...´´*
   was changed

2. *p.2, l.31-33: Why you use only the nominator - the buoyant production of turbulence - of the gradient Richardson number Ri to parametrise tropospheric instabilities and not Ri itself?*
   The standard mixing scheme is driven be the (isentropic) flow deformations and is always active. In particular, in the troposphere, a combination of both the deformation-driven part and of the new part driven by the vertical instabilities is considered. To get more freedom in our parameterisation, we do not combine both effects in only one Richardson number Ri but but keep both parts independent. This is also motivated by the fact that the use of a predefined gradient Richardson number is strongly related to the resolved spatial scales. To keep it open, we use both parts in an independent way. Thus, gradient Richardson number Ri is a heuristic idea which only guides us how to proceed.

3. *p.3, l.34-38: Just for curiosity, would the actual CLaMS transport scheme accumulate tracer in the PBL, if one uses emissions instead of prescribed PBL mixing rations?*
   We've never tried it. CLaMS is a CTM with prescribed boundary conditions in terms of

prescribed mixing ratios. If fluxes are prescribed it is not clear if such fluxes are consistent with the vertical velocities derived from the reanalysis. Nevertheless, for simple mass conservation experiments, we plan to start with such a setup of the model in the future.

4. *p.4, l.5-7: "Although we are aware of numerous convective schemes (e.g. Tiedtke (1989); Emanuel (1991)), our approach mainly intends to cover the range of possible variability due to unresolved tropospheric transport´´. It is quite unclear to me what is meant here, see also point 1.) in the general comments.*
We completely reformulated this paragraph
P4 L10-16

5. *p.5, Figure 2: It seems that the TIL in the extratropics is (very) weakly pronounced compared to e.g. Birner et al. (2006). Is there an explanation? Birner, T., D. Sankey, and T. G. Shepherd (2006), The tropopause inversion layer in models and analyses, Geophys. Res. Lett., 33(14), doi:Doi 10.1029/2006gl026549.*
In the revised version, we explained this point more carefully
P6 3-6

6. *p.7, l.23-26: How the free parameter Nc, the critical limit for static stability, below which enhanced tropospheric mixing is triggered in the new parameterisation, is estimated? Which $N_c$ value has been used for the CLaMS control simulations? How sensitive are the results to the choice of $N_c$?*
$N_c^2$ is set $1.10^{-4}$ ($1/s^2$). This value is now explicitly given in caption of Table 1. Furthermore, we decided to shift all sensitivity studies to the next paper (see our concluding remarks in the last section of the paper). There are also other parameters which have to by analysed. For this we would like to use tropopause-based coordinated and maybe more experimental data

7. *p.8, l.1-2: I understand, that it is technically easier and better comparable to the previous CLaMS version, if the step of the additional tropospheric mixing is executed after the adaptive regridding, but does this also makes sense from a physical point of view? Would the result be different with the inverse transport operator: First step: mixing due to vertical instability and second step: mixing due to strong horizontal wind shear (deformation)?*
Yes, we agree that there is this type of freedom which should be analysed in the future. This is also related to our answer to specific comment 2.

8. *p.9, Figure 4: A similar question: Why you implemented the vertical displacement in your new parameterisation for unresolved deep convection after the horizontal displacement? Is this realistic for deep convection?*
We think that this is a misunderstanding of Fig. 4b. For 6h trajectories, the convective uplift of trajectory may happen either at the beginning or at the and of the trajectory step. We tried now to make it more clear in the caption of Fig. 4b as well as in the main text.

9. *p.9, l.11: Same question as for Nc: How sensitive are the results to the choice of the criteria $N_m^2 < 0$ triggering convective events in the parameterisation?*
See our answer to your specific comment 6

10. *p.11, l.1-2: "It means that the mixing procedure is able to adjust a certain increase or decrease in the number of air parcels, but this amount should be below $\pm 10\%$". This has to be better explained, see point 1.) in the general comments.*
    This is related to our default mixing scheme, in particular, how often we run it to "put together" all the air parcels which are too close to each other (and it happens, if too many air parcels are lifted by convection). This also a point, which we would like to handle more carefully in the future

11. *p.11, l.1-2: Again, how sensitive are the results of the control simulations to the choice of criteria that only $\theta > 35$ K triggers convection? Would it not be better to use instead of a fix value the criteria $theta + \Delta\theta >$ upper level of the PBL? This would mean that only convective events are considered that lift the air parcel out of the PBL, where the CO2 mixing ratios of the air parcel will be overwritten by the prescribed value in the next time step anyway.*
    In our convection updrafts we consider only air parcels within the lowest model layer which approximates the PBL. The lowest layer (i.e. the PBL) is then overwritten by the prescribed boundary condition. Thus, in these two points (only air parcels from the PBL are lifted and CO2 in the PBL is always prescribed) we exactly follow the recommendations. Concerning the sensitivity to the used parameter, we would like to do in the future, in particular in connection with the correction of the mass conservation as we discuss it in our completely revised section 3.3
    P9 L1 - 13, P10 L27 - P12 L33

12. p.12, l.17-20: *"It should be emphasized..." What would this mean for simulations using emission fluxes rather than prescribed surface mixing ratios (see also my question p.3, l.34-38 above)?*
    See our answer of your specific comment 3

13. *p.13, Table 1: The critical values of the dry and moist Brunt Vaisala frequency used for the simulations should be added here. Also, it would be easier to use different names for both simulations FULL_EXT (e.g. FULL_EXT_0.3 and _0.7).*
    Here, we completely followed your recommendations

14. *p.14, Figure 8: Please specify the simulations: Top: FULL_EXT_0.3 or FULL_EXT_0.7? Bottom: REF or REF-6h?*
    was done

15. *p.15, l.4: Boucher et al. (2009) is not really a good citation for stratospheric CO2 and methane oxidation. The chemistry of methane in the middle atmosphere was to my knowledge first considered by Bates and Nicolet (1950). Early measurements of stratospheric CO2 and CH4 profiles date back to the 1960s and 70s, e.g. by Ehhalt. Bates, R. D., and M. Nicolet (1950), Atmospheric Hydrogen, Publications of the Astronomical Society of the Pacific, 62(365), 106. Ehhalt, D.H., The atmospheric cycle of methane. Tellus: 26, 58, 1974. Ehhalt, D.H., L.E. Heidt, R.H. Lueb, and E.A. Martell, Concentrations of CH4, CO, CO2, H2, H2O and N2O in the upper stratosphere. J Atmos Sci: 32, 163, 1975.*
    Thanks a lot for these citations, so we included 2 of them

16. *p.15, l.16: How the reference simulations has been initialized on 1.1.2000? With an empty atmospheric domain - no CO2 ? If so, the spin-up of only 5 years might be a bit too short for UTLS CO2 analysis.*
    Both reviewers asked for more details describing the details of our runs. For this we put this information into our revised text: see first paragraphs in section 3.1 and 4.1

17. *p.15, l.16: "... all other control runs were started using the output of the reference run for the initial distribution.´´ This is a bit unclear to me. Does this mean that the control runs have started with the output of the reference run from 31.12.2004? If so, the results of the control runs for 2005, at least for the LS, is influenced by the prescribed distribution of the reference simulation. In the UT, this should only influence the first months of 2015, but still, this is not optimal for the comparison.*
    see the above answer

18. *p.15, l.25: "...on, the former being in better agreement with CarbonTracker´´. I think, this statement is a bit misleading, because CO2 assimilated from CarbonTracker data set is not the reference for the middle troposphere or UTLS, see also general comment point 2.).*
    Yes, we agree, so we removed the second part of the sentence.

19. *p.17, Figure 10: Should be improved, see general comment point 2.).*
    Was partially done by including the standard deviations. See also our answer to your main second point.

20. *p.17, Figure 10, Legend: According to Table 1, the name of the data set in the legend should be TROP_MIX and not VERT_MIX.*
    We adjusted all abbreviations following the recommendation

21. *p.17, Figure 10, Caption: The altitude range in (b) is 10.5 km and not 15.5 km.*
    was changed

22. p.17, l.3-5: For the extratropics the gridding of the aircraft data has to be done in tropopause related coordinates or has to be filtered for tropospheric data, see general comment point 2.). Otherwise the averages are strongly biased by the sampling statistic of tropospheric and stratospheric air in the individual bins.
    See also our answer to your main second point.

23. *p.17, l.5-6: Has CLaMS simulations been sampled along the flight track?*
    No, we use the same type of climatologies as described in Diallo et al., ACP, 2017. See also our answer to your main second point.

24. *Sec.4.4, p.18, l.10-11: This should be simply demonstrated by comparing REF vs. REF-6h.*
    was done

25. *Sec.4.4, p.18, l.16-18: This might be true, but I cannot really see the differences in AoA between Fig 11b and c. To demonstrate this, a difference plot would be necessary.*
    Figure 11 was completely changed, so we show now the differences of AoA and not the absolute values. In this way, the reader can see step by step (Fig. 11b to f), how our changes of tropospheric transport can be seen in the AoA distribution. The related subsection 4.4 was

completely rewritten.
P20 L8 - P21 L17

[revised manuscript text omitted]

2, 153–173, 2009.

---

## Author Response (AR2)

Response to Astrid Kerkweg

First we apologize for not answering your comments in our first response (we thought that we addressed all these points during the first technical revision where we followed the recommendations from Volker Grewe which were very similar to your comments). In the following, we address all the points raised in your review: Respecitve changes in the manuscript are highlighted in red.

1. *Therefore please add the version number of CLAMS in the title of your publication upon revision.*
   ...was done, see title and abstract. The new version is 2.0, the old version 1.0 and was described in Pommrich et al., 2014. Both version numbers are clearly denoted in the manuscript.

2. *Additionally, the code availability section is really confusing. So what is your publication about? Did you used CLAMS or CLAMS/MESSy? Please note, that the Code Availability Section primarily refers to the exact version, which is published in the article. So if you used CLAMS (without MESSy) so please state how to access CLAMS. If you used CLAMS/MESSy state it in the article itself. Of course, you are free to add references the further developed version (here CLAMS/MESSy) in the Code Availability section. In that case, please state in which version of MESSy the new updates will be available and please add more information on how to access MESSy and especially CLAMS/MESSy.*
   The Code Availability Section was rewritten. Thus, both versions (CLaMS 2.0 and CLaMS 1.0) can now be downloaded from a GitLab server. Furthermore, CLaMS 2.0 has also a DOI: 10.5281/zenodo.2632683.

[revised manuscript text omitted]